



# The JUICE 2024 close flyby of the Moon: Thermal assessment from MAJIS

Federico Tosi[1], Clément Royer[2], Federico Colaiuta[3,1], François Poulet[2], Tyler M. Powell[4], Benjamin T. Greenhagen[4], Yves Langevin[2], Alessandro Mura[1], Giuseppe Piccioni[1], Cédric Pilorget[2], Cristian Carli[1], Francesca Zambon[1]

[1]Istituto Nazionale di Astrofisica – Istituto di Astrofisica e Planetologia Spaziali (INAF–IAPS), Rome, Italy
[2]Institut d'Astrophysique Spatiale, CNRS/Paris-Saclay University, Paris, France
[3]University of Rome "La Sapienza", Department of Physics, Rome, Italy
[4]Johns Hopkins University – Applied Physics Laboratory (JHU–APL), Laurel (MD), USA

*Correspondence to*: Federico Tosi (federico.tosi@inaf.it)

**Abstract.** During the August 2024 lunar flyby of the Jupiter Icy Moons Explorer (JUICE), the MAJIS imaging spectrometer acquired the first hyperspectral observations of the Moon extending up to 5.56 µm at sub-kilometre resolution. This dataset provides an unprecedented opportunity to investigate the near-infrared thermal emission of the lunar surface and to validate MAJIS capabilities in a well-characterized planetary environment. We derive surface temperature and spectral emissivity using three independent approaches: a Bayesian inversion constrained by radiative transfer, an empirical correction based on laboratory relationships for lunar soils, and a roughness-informed thermal model that explicitly accounts for surface geometry and anisothermality. All methods reproduce the expected dependence of temperature on solar illumination, while their divergences at high incidence angles highlight the role of roughness and unresolved topography. The roughness-informed model achieves the closest agreement with thermophysical predictions, whereas the Bayesian and empirical approaches exhibit complementary strengths under different illumination regimes. Emissivity retrievals consistently reveal higher values in mare regions than in surrounding highlands, reflecting known compositional and textural contrasts, and show a wavelength-dependent inversion relative to longer-wavelength Diviner measurements. These results establish a validated framework for MAJIS thermal analysis of airless bodies and provide a benchmark for its future application to the investigation of the Jovian satellites.

## 1 Introduction

The European Space Agency's JUpiter ICy moons Explorer (JUICE) mission (Grasset et al., 2013), launched in April 2023, executed a close lunar flyby in August 2024 as part of its interplanetary trajectory toward Jupiter. While JUICE is optimized for the exploration of Jupiter's icy satellites, this flyby provided a critical opportunity to validate the instrumental performances of the scientific payload within a well-characterized planetary environment. For the Moons and Jupiter Imaging Spectrometer (MAJIS) (Poulet et al., 2024), this flyby served a dual purpose: validating MAJIS performance and demonstrating its capability



to retrieve physical and compositional surface properties under known lunar conditions at sub-kilometre spatial resolution and with unprecedented spectral resolution (Poulet et al., this issue).

Characterizing the Moon's surface thermal environment has been a long-standing scientific objective. Early ground-based and Earth-orbital telescopic observations provided bulk thermal properties and large-scale diurnal trends. The Apollo program acquired the first in situ heat flow measurements (e.g., Apollo 15, 17) and direct surface temperature records (Keihm et al., 1973; Keihm and Langseth, 1975). However, high-resolution orbital thermal mapping of the Moon only became routine with the Diviner Lunar Radiometer Experiment aboard NASA's Lunar Reconnaissance Orbiter (LRO) (Paige et al., 2010). Since 2009, Diviner has delivered near-global, multi-annual coverage of lunar surface temperatures at ~200 m/pixel resolution across multiple local times, enabling detailed studies of thermal inertia, rock abundance, rock size-frequency distributions, and volatile stability in permanently shadowed regions. Despite this revolutionary dataset, a significant observational gap remains in the near-infrared range 3.0–5.5 μm, which captures the Wien tail of the thermal emission and provides sensitivity to surface emissivity variations linked to regolith texture, roughness, and composition. This interval thus complements the longer-wavelength coverage of Diviner, extending high-resolution thermal measurements into a spectral domain previously unexplored at global scales.

Prior to JUICE, the Moon had been observed by other imaging spectrometers, although none provided the combination of spatial resolution and spectral coverage needed for detailed thermal analysis. The Moon Mineralogy Mapper ($M^3$) onboard Chandrayaan-1 (Pieters et al., 2009) achieved near-global mineralogical mapping with a typical spatial resolution of 140 m/pixel but was limited to wavelengths shorter than 3 μm. The Visible and Infrared Mapping Spectrometer (VIMS) aboard Cassini (Brown et al., 2004) observed the Moon during its Earth swing-by in August 1999, mainly for the purpose of calibration. While VIMS covered wavelengths up to 5.1 μm, the data acquired during the lunar flyby suffered from both very coarse spatial resolution (∼192 km/px) and severe saturation above 3.7 μm (Bellucci et al., 2002), which together precluded their use for any meaningful thermal mapping. Likewise, the JIRAM spectro-imager onboard NASA's Juno spacecraft (Adriani et al., 2017) acquired images and spectra of the Moon during the Earth flyby that occurred in October 2013, but with limited spatial resolution (52–55 km/pixel), which enabled thermal retrieval in the range 3.0–4.2 μm mostly for the purpose of validating the instrument performances (Adriani et al., 2016). More recently, the Imaging Infrared Spectrometer (IIRS) on board Chandrayaan-2 (Chowdhury et al., 2020) acquired hyperspectral data of the Moon in the 0.8–5.0 μm range at a spatial resolution of approximately 80 m/pixel, demonstrating its capability to retrieve lunar surface temperatures and emissivity in the 3.0–5.0 μm range at local scale (Verma et al., 2022; Ojha et al., 2024). MAJIS observations during the JUICE flyby provide a complementary dataset characterized by a distinct illumination geometry, a broader spectral range extending to 5.56 μm, and a different instrumental heritage, allowing for independent validation of retrieval methods and offering new insights into the thermophysical properties of specific lunar regions under varying solar illumination.

On August 19, 2024, during the JUICE Lunar Gravity Assist (LGA), in the outbound leg of the flyby shortly after closest approach, MAJIS acquired four hyperspectral images of the lunar surface in the overall spectral range 0.49–5.56 μm. We tag these observations respectively: C1, C2, C3 and C4. Taken from altitudes of 874 to 2406 km over the surface, these datasets





yielded average pixel scales between 0.13 and 0.36 km/pixel, providing exceptional sub-kilometre infrared coverage across varying incidence angles and local times (a detailed description of the MAJIS data and the flyby geometry is documented in Poulet et al., this issue). Table 1 summarizes key specifications of the MAJIS LGA data. In the C4 observation, the first 17 out of 64 pixels along the slit ("*samples*") are not usable because the observation was purposely commanded to illuminate only

part of the slit to evaluate the magnitude of straylight; therefore, they are ignored in the subsequent thermal analysis.

**Table 1.** Main features of the four hyperspectral images acquired by MAJIS during the JUICE flyby of the Moon on August 19, 2024.

| | C1 | C2 | C3 | C4 |
|---|---|---|---|---|
| **Filename** | 20240819211816 | 20240819211923 | 20240819212141 | 20240819212402 |
| **Start time (UTC)** | 2024-08-19T21:18:16 | 2024-08-19T21:19:24 | 2024-08-19T21:21:42 | 2024-08-19T21:24:02 |
| **Stop time (UTC)** | 2024-08-19T21:19:06 | 2024-08-19T21:21:31 | 2024-08-19T21:23:49 | 2024-08-19T21:25:26 |
| **Size (samples × lines × bands)** | 400 × 99 × 1016 | 64 × 1274 × 1016 | 64 × 1269 × 1016 | 64 × 841 × 1016 |
| **Altitude over the surface (km)** | 873.95 – 942.45 | 971.52 – 1269.24 | 1286.80 – 1813.27 | 1791.97 – 2406.05 |
| **Pixel resolution (km)** | 0.131 – 0.141 | 0.146 – 0.190 | 0.193 – 0.272 | 0.269 – 0.361 |
| **Phase angle (deg)** | 88.5 – 91.9 | 89.9 – 90.5 | 89.9 – 90.5 | 88.4 – 88.9 |
| **Solar incidence angle (deg)** | 85.0 – 93.1 | 66.2 – 83.7 | 44.4 – 65.2 | 26.3 – 45.1 |
| **Emission angle (deg)** | 0.2 – 7.0 | 8.2 – 26.0 | 25.2 – 47.9 | 43.4 – 65.7 |
| **Local solar time (h)** | 17.6 – 18.2 | 16.4 – 17.6 | 15.0 – 16.3 | 13.7 – 15.0 |

The JUICE/MAJIS flyby data complement the legacy of Diviner in two key ways: (1) they achieve spatial resolution comparable to Diviner during a targeted campaign, and (2) they extend high-resolution coverage into the 3.0–5.56 μm infrared spectral domain, which is under-explored for the Moon. Unlike Diviner's broadband and multispectral thermal infrared (TIR) channels, MAJIS provides hyperspectral coverage across the 2.28–5.56 μm range with an average ~7-nm spectral resolution (Haffoud et al., 2024). This interval captures the Wien tail of thermal emission and may include weak overtone or combination

features of silicate materials. The hyperspectral capability enables the simultaneous retrieval of brightness temperature and spectral emissivity with high fidelity. Such data reveal subtle variations diagnostic of surface properties including roughness, regolith grain size, porosity, and composition—parameters that are often difficult to disentangle using broadband measurements.

Making the most of this small but unique dataset, the primary scientific objectives of the MAJIS lunar observations are: (1) to

map diurnal temperature variability at high spatial resolution, (2) to retrieve spectral emissivity longward of 3 μm, and (3) to assess thermophysical properties where constrained by observation geometry. While the single flyby precludes broad-scale



thermal inertia mapping, the hyperspectral data enable targeted thermophysical insights at specific locations. Localized modelling—leveraging distinct observation geometries, known crater morphology, and predicted self-heating effects—allows quantification of surface roughness at sub-kilometre scales. Furthermore, spectral emissivity retrievals provide independent constraints on regolith texture (grain size, packing) intrinsically linked to thermophysical behaviour. These point estimates offer valuable validation against Diviner regional trends and in situ data.

These targeted results advance understanding of small-scale lunar surface processes while serving as a critical proving ground for MAJIS methodologies. Quantifying roughness and texture under known lunar conditions can help validate techniques for interpreting thermal data from Ganymede, Callisto, and Europa, where similar sunlit observations by MAJIS are planned. The flyby thus directly bridges lunar science and JUICE's core exploration goals.

In the remainder of this paper, Section 2 presents the derivation of surface temperature and emissivity from MAJIS data using three independent approaches: (i) a Bayesian nonlinear inversion constrained by radiative transfer, (ii) an empirical thermal correction adapted from laboratory-based relationships, and (iii) a roughness-informed, physically consistent thermal model. Before intercomparing these methods, we introduce a cross-wavelength context by incorporating co-located LRO/Diviner observations to establish an external benchmark for both temperature and emissivity (subsection 2.4). We then evaluate the performance of the three approaches in terms of their ability to reproduce observed thermal behaviour and emissivity contrasts across different terrains (subsection 2.5). Finally, in Section 3 we synthesize these results in a broader scientific discussion, drawing conclusions on the thermophysical properties of the lunar surface and outlining the implications for future applications of MAJIS to airless bodies in the outer Solar System.

## 2 Derivation of surface temperature and emissivity from MAJIS data

### 2.1 Bayesian approach to nonlinear inversion

The Bayesian nonlinear inversion used for MAJIS temperature retrieval builds upon techniques previously validated across multiple planetary missions. The methodology was routinely used on data acquired by the Dawn/VIR imaging spectrometer, which achieved nearly global coverage at asteroid Vesta and dwarf planet Ceres, leveraging the nonlinear radiance-temperature relationship to infer thermophysical properties while accommodating rapid rotational cycles (Tosi et al., 2014; Capria et al., 2014; Tosi et al., 2015; Tosi et al., 2018). The methodology's adaptability was further demonstrated on infrared data acquired by Rosetta/VIRTIS at the asteroid Lutetia (Keihm et al., 2012) and especially at the nucleus of comet 67P/Churyumov–Gerasimenko, resolving self-heating phenomena within shadowed concavities and diurnal thermal gradients at 15 m/pixel resolution (Tosi et al., 2019). Most directly, a similar Bayesian workflow was applied to JIRAM data of the Moon acquired during the 2013 Earth flyby of Juno to derive temperatures validated against LRO/Diviner data (Adriani et al., 2016).

The details of the Bayesian approach to nonlinear inversion are explained in the Appendix A of Tosi et al. (2014). Briefly, this technique addresses the inherent challenge of solving for multiple unknowns—temperature and wavelength-dependent





emissivity—from a limited set of spectral radiance measurements. For a retrieval involving $N$ spectral channels, the state vector includes $N$ emissivity values plus the surface temperature, for a total of $N + 1$ free parameters.

The method follows the classical maximum-a-posteriori (MAP) formulation of Rodgers (2000), solved iteratively using a Gauss–Newton scheme. The radiative transfer model is linearized around an initial a priori state, and the optimal solution is obtained by minimizing a quadratic cost function combining the spectral misfit and the prior constraints. The associated posterior covariance matrix—derived from the inverse of the approximate Hessian of the cost function—provides the formal uncertainties of the retrieved parameters.

The physical basis of the approach lies in the radiance equation for airless bodies, where the measured spectrum combines reflected solar and thermally emitted radiation. Kirchhoff's law ($r = 1 - \varepsilon$) links reflectance to emissivity, but the system remains underconstrained without additional prior information. The inversion therefore begins with an estimate of surface temperature obtained from the brightness temperature in the dominant thermal-emission region, assuming an initial constant emissivity. An important refinement is the dynamic determination of the crossover wavelength—the point at which reflected

and emitted radiation are equal—based on this temperature estimate. The spectral domain used in the inversion is then defined as extending from 0.5 μm shortward of this crossover to the long-wavelength limit of MAJIS sensitivity, ensuring that the crossover region is always included for a stable emissivity solution.

Priors constrain the physically meaningful variability of the unknowns (±30 K for temperature, and wavelength-dependent bounds on emissivity to ensure $\varepsilon \leq 1$ and avoid nonphysical solutions). In practice, these values represent soft constraints: they define the width of the a priori covariance matrix but do not impose hard limits. The Gauss–Newton iterations may therefore

converge outside these ranges if supported by the radiance data. Only two hard constraints are applied. First, emissivity values exceeding unity are clipped to $\varepsilon = 1$. Second, spectral segments are discarded when the Bayesian emissivity differs from the Kirchhoff-derived estimate by more than 4%, a criterion that removes artifacts near the crossover region. These steps prevent non-physical behaviour while preserving the flexibility inherent in the Bayesian formulation. Formal uncertainties of

temperature and emissivity are computed from the posterior covariance and incorporate the in-flight Noise Equivalent Spectral Radiance (NESR), which increases toward longer wavelengths due to spectrometer background.

A Bayesian formulation offers significant advantages over a simple least-squares fit when estimating temperatures from infrared spectra, particularly when the available wavelengths lie in the Wien tail of the Planck function. In this regime, radiance depends exponentially on temperature, creating a strong degeneracy between temperature and emissivity and making least-

squares solutions highly sensitive to noise and to the choice of initial emissivity. Instead of relying on a scalar emissivity fitted simultaneously with $T$, the MAP framework adopted here uses a full covariance matrix of the unknowns to regularize this degeneracy. Although the Gauss–Newton implementation retrieves only the local posterior maximum rather than the full posterior distribution, the posterior covariance naturally provides formal uncertainties that reflect the structure of the inverse problem. In particular, spectral covariance imposes smoothness across adjacent wavelengths—consistent with the behaviour

of natural granular materials—while allowing physically plausible variability.





Compared to other methods that fit a Planck function with a single emissivity value, the Bayesian retrieval exploits the full covariance matrix of the unknowns as a genuine a priori. This matrix encodes expectations about how emissivity varies with wavelength, enforcing smoothness through statistical correlations between neighbouring spectral channels—typically modelled with Gaussian decay over a correlation length comparable to the spectral resolution. This suppresses high-frequency

noise and reflects the physical reality that natural surfaces lack abrupt spectral features. In regions where radiance data are ambiguous—such as near the crossover between reflected sunlight and thermal emission—the covariance matrix stabilizes the solution, preventing unrealistic oscillations. It also governs the formal uncertainties, balancing constraint and flexibility: tighter correlations reduce noise but risk over-regularization, while looser ones capture more variability at the expense of stability. The result is a spectral regularizer rooted in physical smoothness rather than mineralogical assumptions.

Subsequent iterations refine temperature and emissivity simultaneously. For each pixel, the algorithm iterates up to 50 times, recalculating the Jacobian matrix (radiance sensitivity to parameter changes) at each step to update the solution and its uncertainties. Post-retrieval validation removes emissivity data affected by saturation or inconsistent with the Bayesian solution, ensuring robustness. The method achieves convergence even with suboptimal initial guesses, yielding reliable results for temperatures >170 K (the exact limit depends on sensitivity range and instrument thermal conditions) for dayside

measurements. Typical formal uncertainties are 1–3 K for temperature in well lit areas and 0.01–0.20 for emissivity, increasing for low-temperature and/or high incidence angle scenarios.

For MAJIS, the heritage methodology was systematically adapted to lunar conditions. First, we consider data acquired only in the infrared (IR) channel, which covers the spectral range 2.28–5.56 µm. Raw data underwent reinsertion of the background signal (which is automatically subtracted in the calibration pipeline) to identify saturation artifacts, which appear as plateaus

in the spectral profile, sometimes with oscillations. Spectral pixels ("*spectels*") exceeding 10,400 Digital Numbers (DN)—a threshold empirically derived from C4 data, which experience the highest solar illumination—were flagged and excluded. Synthetic radiance spectra were generated by combining thermal emission with reflected solar components modelled using the MODTRAN Cebula+Kurucz extraterrestrial solar spectrum (https://www.nrel.gov/grid/solar-resource/spectra.html) scaled for the heliocentric distance of the Moon at the time of the observation (1.014 AU). Several hypotheses on the initial emissivity

(0.70–0.95) were evaluated, and the initial temperature was set equal to the brightness temperature in the 5.27–5.56 µm range, allowing both parameters to vary within noise constraints until convergence. Two spectral ranges were tested: the computationally efficient 4.5–5.56 µm window (Table A1) and the broader 3.0–5.56 µm interval (Table A2), the latter reducing saturation biases in high-temperature regions at the cost of increased processing load. The Lommel–Seeliger disk function (Hapke, 1981)—a photometric law optimized for the Moon—was also tested to account for illumination and viewing geometry

(Table A3).

Geometric precision was ensured through SPICE kernel navigation data (Acton, 1996; Acton et al., 2018) and instrument kernels, defining ellipsoid-level planetocentric coordinates as well as solar incidence and emission angles. The LEGA observations enable an update of the MAJIS instrument kernel, with details provided in a companion paper dedicated to the geometric calibration (Seignovert et al., this issue). Topographic effects may be mitigated using a digital shape model to



compute local illumination and emission angles. However, while such corrections are critical for irregularly shaped bodies such as asteroids and comets, as well as for high-resolution optical imagery in general, approximating the Moon's shape as a smooth ellipsoid is acceptable in the case of MAJIS data, whose pixel scale is on the order of a few hundred meters. Emissivity outputs were restricted to the spectral range used for temperature retrieval (4.5–5.56 µm and 3.0–5.56 µm).

Using the range 4.5–5.56 µm, the temperature values obtained through the Bayesian inversion across the four MAJIS lunar

observations (C1 to C4) reveal systematic dependencies on both the level of solar illumination and the choice of prior emissivity. As expected, the mean surface temperature increases from C1 through C4, reflecting progressively stronger solar heating (Figures 1–2). C1, acquired under grazing illumination, shows the lowest mean temperatures (~176–184 K), while C4, obtained under near-equatorial midday conditions, yields mean temperatures approaching 356–343 K, depending on the emissivity prior (Figure 3 and Table A1). A representative example of the spectral fit and the associated posterior temperature

distribution for an individual MAJIS pixel is shown in Figure A1, illustrating the behaviour of the Bayesian inversion described above.



**Figure 1. Bayesian temperature retrievals (C1–2).** (**a**) Temperature maps derived from MAJIS observations C1 and C2, using the Bayesian approach applied to the 4.5–5.56 µm portion of the IR data using an emissivity prior of $0.70 \pm 0.15$ and no photometric correction. C1 is the easternmost data, taken near the terminator. C2 extends from northwest to southeast at low southern latitudes. Both data cover lunar highlands. Pixel resolution goes from 0.19 to 0.36 km. Colour gradation is mainly due to instantaneous solar illumination combined with local topography. (**b**) Formal uncertainties associated with the retrieved temperature values. The largest uncertainties are associated with the lowest temperature values, corresponding to unlit locations. Background image: Moon LRO LROC WAC Global Morphology Mosaic 100m, June 2013 (Speyerer et al., 2011; Wagner et al., 2015).





**Figure 2. Bayesian temperature retrievals (C3–4).** (**a**) Temperature maps derived from MAJIS observations C3 and C4, using the Bayesian approach applied to the 4.5–5.56 μm portion of the IR data under an emissivity prior of $0.70 \pm 0.15$ and no photometric correction. C3 is the easternmost data and mostly covers Mare Fecunditatis, bordering Langrenus crater in the southeastern edge. C4 is the westernmost data ad embraces Mare Tranquillitatis and highlands in between Tranquillitatis and Fecunditatis. Both data extend from northwest to southeast at low southern latitudes, crossing the equator. Pixel resolution goes from 0.13 to 0.19 km. The colour gradation reaches its maximum on the scale, about 380 K, at the westernmost points of C4, which experience early afternoon. (**b**) Formal uncertainties associated with the retrieved temperature values. Typical average uncertainty is ~1 K. Background image: Moon LRO LROC WAC Global Morphology Mosaic 100m, June 2013 (Speyerer et al., 2011; Wagner et al., 2015).



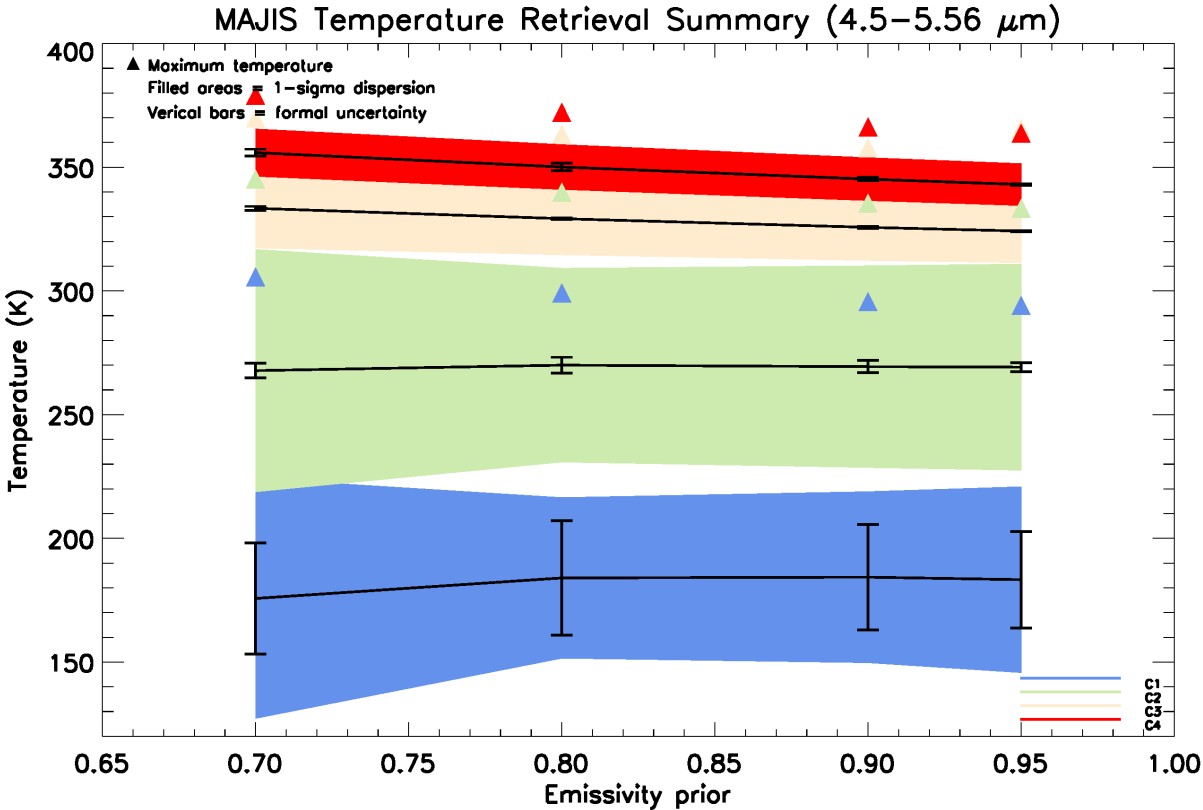

**Figure 3. Summary of MAJIS temperature retrievals as a function of emissivity prior (4.5–5.56 µm spectral range).** The coloured bands represent the retrieved mean surface temperature (solid lines) and the corresponding $\pm1\sigma$ dispersion for the four MAJIS observations (C1–C4). Vertical black bars indicate the formal uncertainties associated with the Bayesian inversion, while triangles mark the maximum temperature values observed within each dataset. The results illustrate the sensitivity of the retrieved temperature to the assumed emissivity: for the warmer datasets (C3 and C4), the mean temperature decreases slightly with increasing emissivity prior, while for the colder, high incidence cases (C1 and C2) the dependence on the emissivity prior is weak and does not follow a strictly monotonic trend; small, irregular variations with prior reflect low thermal signal, local illumination/topography and retrieval noise rather than a systematic physical response.

As expected, the choice of initial emissivity ($\varepsilon_0$) influences the retrieved temperatures. Lower assumed emissivity values yield higher temperature estimates, due to the inverse relationship between emissivity and brightness temperature in the Planck function. In high-SNR observations such as C4, this leads to differences in mean temperature as large as 13 K across the range of tested priors. In contrast, datasets acquired at higher incidence angles (e.g., C1 and C2) exhibit smaller temperature shifts across priors, likely because shadowed regions dominate the radiance signal, flattening the temperature distribution and dampening the sensitivity to $\varepsilon_0$.





The uncertainties associated with the retrieved temperatures show a more pronounced dependence on both solar illumination and the width of the emissivity prior. Formal uncertainties and $1\sigma$ dispersions are significantly higher in C1 and C2, particularly when using broad priors (e.g., $\varepsilon_0 = 0.70 \pm 0.15$), reflecting the limited thermal contrast and stronger model degeneracy under

low radiance conditions. In contrast, observations with higher insolation—namely C3 and C4—yield markedly lower formal errors and temperature dispersion. This reduction in spread is not only a function of radiometric quality but also of scene illumination: in C4, the entire observed area is well lit, and the few remaining shadows are due solely to local topography. This homogeneity in thermal forcing minimizes temperature variability across the scene and enhances retrieval precision, reinforcing the robustness of the Bayesian approach under favourable conditions.


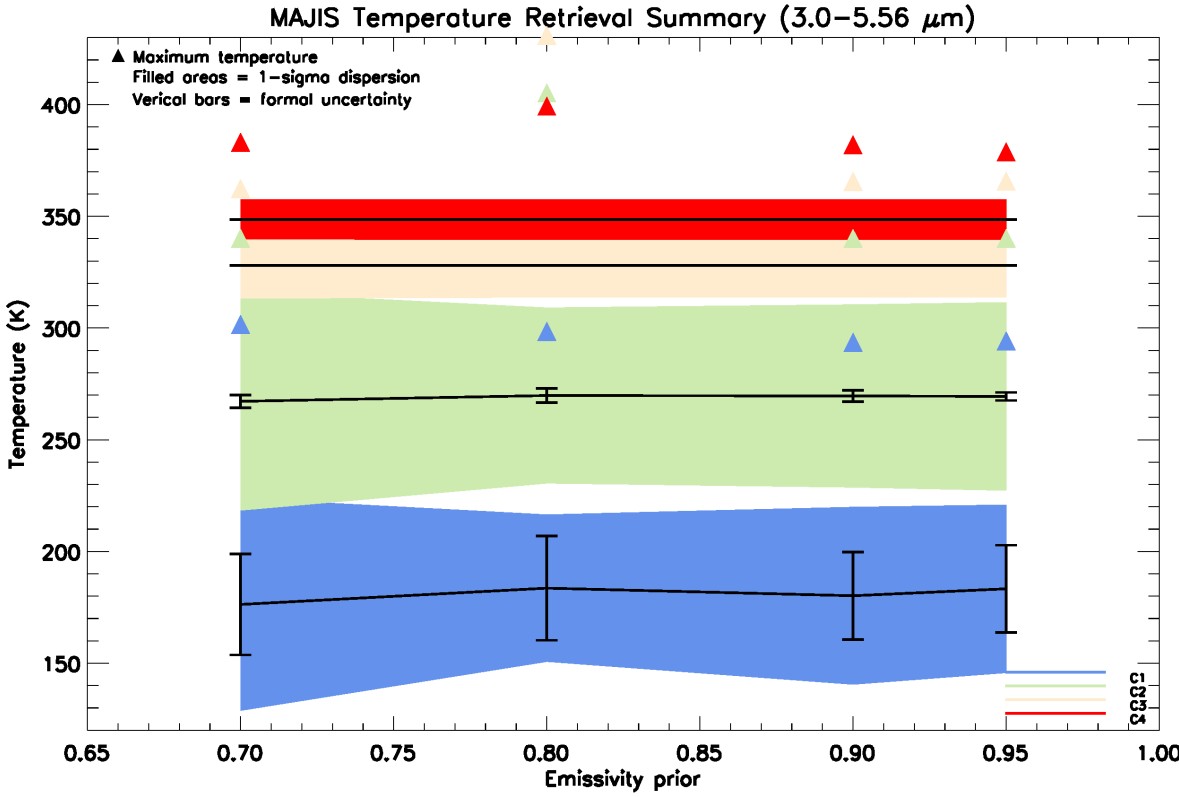

**Figure 4. Summary of MAJIS temperature retrievals as a function of emissivity prior (3.0–5.56 µm spectral range).** The coloured bands show the mean surface temperatures (solid lines) and corresponding $\pm 1\sigma$ dispersion for the four MAJIS observations (C1–C4), obtained from Bayesian inversion without photometric correction. Vertical black bars represent the mean formal uncertainties, and triangles

indicate the maximum retrieved temperature values. Compared with the 4.5–5.56 µm case, the wider spectral interval yields similar temperature trends but generally lower formal uncertainties, especially under well illuminated conditions.





Results show that extending the lower bound of the spectral window from 4.5 to 3.0 μm produces a significant change in the
character of the inversion (Figure 4 and Table A2). When using the full 3.0–5.56 μm range, the retrieval yields temperature
values that are nearly invariant with respect to the emissivity prior, especially in the well illuminated datasets C3 and C4. In
these cases, the mean surface temperatures converge to a fixed value regardless of the prior, and, while the associated 1$\sigma$
dispersion is substantially similar to the 4.5–5.56 μm case, the formal uncertainties collapse to zero, indicating that the
inversion algorithm has converged to a narrow solution space (Figure 5). Although such tight convergence might suggest
improved precision, it more likely reflects numerical over-constraint. This behaviour is consistent with an over-dominant
contribution from shorter wavelengths—between 3.0 and 4.2 μm—where thermal emission is still weak and the radiance
spectrum is more sensitive to the shape of the assumed reflectance curve and potential calibration uncertainties.

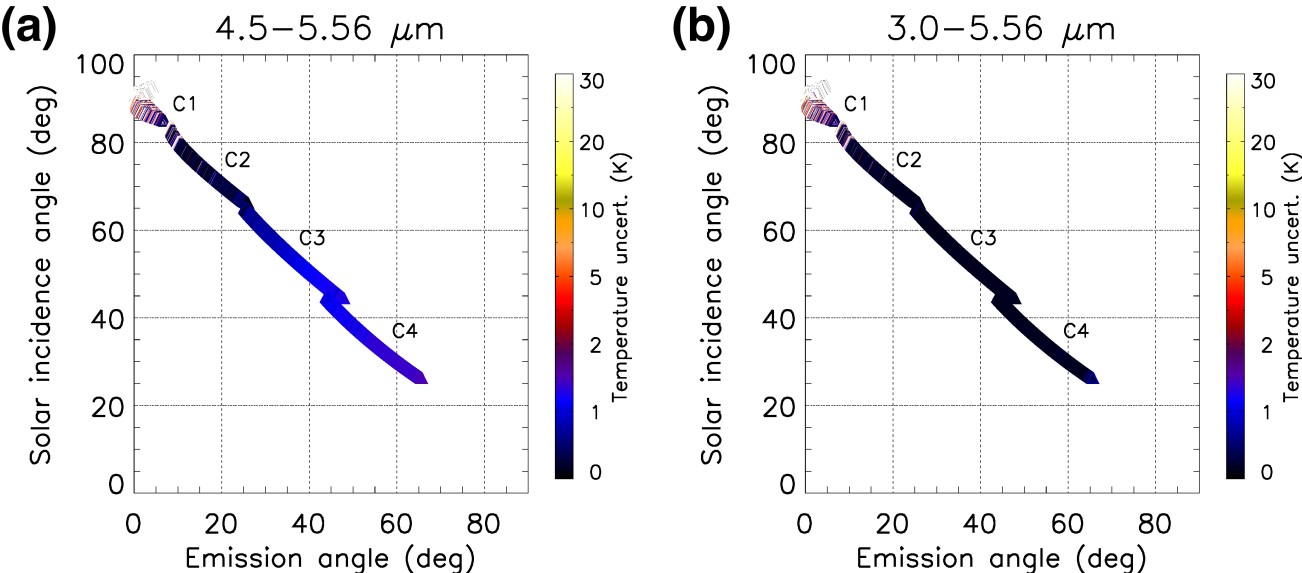

**Figure 5. Formal uncertainties vs geometry.** Formal uncertainties associated with the temperature values calculated in the four MAJIS
acquisitions, as a function of the solar incidence angle (*y*-axis) and the emission angle (*x*-axis). Panels (**a**) and (**b**) refer to the retrieval
cases in the range 4.5–5.56 μm and 3.0–5.56 μm, respectively. It can be noted that the wider spectral range results in a lower formal
uncertainty under the same illumination and observation conditions for those data that benefit from full solar illumination.

Comparing corresponding rows across Tables A1 and A2 shows systematic differences in mean temperatures. For example,
in C4 under $\varepsilon_0 = 0.90 \pm 0.10$, the mean temperature increases from 345.2 K (4.5–5.56 μm) to 348.6 K (3.0–5.56 μm), while in
C1 it decreases from 184.3 K (4.5–5.56 μm) to 180.2 K (3.0–5.56 μm). These small shifts reflect the different spectral leverage
of the two wavelength ranges, as shorter wavelengths provide enhanced sensitivity to the thermal continuum at higher
temperatures. In warm, well illuminated conditions, the additional thermal signal from the 3.0–4.5 μm interval strengthens the





fit to the observed radiance, improving the stability of the retrieval where the signal in the 4.5–5.56 μm range alone would begin to saturate.

Despite these potential advantages, the retrieval from the broader 3.0–5.56 μm range may occasionally yield physically implausible outliers. For instance, in C3, a peak temperature of 430.9 K is reported under $\varepsilon_0 = 0.80$—a value well above what is expected for lunar surface conditions at that incidence angle. This suggests that residual artifacts or noise in the shortwave

portion of the spectrum may corrupt the inversion under some conditions. These effects are not observed in the results from the 4.5–5.56 μm range, where temperature statistics behave smoothly and remain consistent with physical expectations.

Overall, these findings indicate that while the inclusion of wavelengths between 3.0 and 4.5 μm can, in principle, enhance retrievals in low-radiance scenes by increasing the spectral baseline, this benefit is offset by increased sensitivity to calibration uncertainties and reflectance modelling. In high-SNR, thermally bright observations such as C3 and C4, the narrower 4.5–

5.56 μm range appears to offer more stable and physically reliable results. It minimizes sensitivity to the crossover region and avoids the numerical dominance of channels with mixed solar and thermal contributions.

These considerations support a context-dependent approach to spectral selection: the full 3.0–5.56 μm range may prove useful in cold or poorly illuminated scenes, whereas the more conservative 4.5–5.56 μm window is preferable when strong thermal emission is present. Future improvements in the treatment of solar reflectance and absolute calibration in the shortwave infrared

may enable more confident use of the full range, but under current processing conditions, restricting the inversion to wavelengths dominated by thermal emission yields more robust and interpretable results.

A final test was performed by applying a Lommel–Seeliger photometric correction to the MAJIS spectra in the range 3.0–5.56 μm before retrieval, with the aim of reducing possible biases linked to viewing and illumination geometry. The Lommel–Seeliger law, originally formulated in the late 19th century to describe single scattering from dark planetary surfaces (Lommel,

1887; Seeliger, 1888), has long been applied to lunar observations as a simple and effective photometric correction. Its role was later formalized within the more general radiative transfer framework of Hapke (1981), where it is recognized as a limiting case particularly suited to the Moon's porous regolith under moderate illumination conditions. The Lommel–Seeliger correction was implemented pixel-by-pixel through the factor $\mu/(\mu+\mu_0)$, where $\mu = \cos(i)$ and $\mu_0 = \cos(e)$ being $i$ and $e$ the solar incidence and emission angle, respectively, and normalizing the observed radiance to a canonical geometry ($i = 60°$, $e = 30°$)

reproducing the high phase angle (~90°) of the JUICE lunar flyby. However, it is precisely this extreme phase angle that makes the correction sub-optimal, which explains why the retrieved temperatures (Figure 6 and Table A3), are virtually indistinguishable from those obtained without photometric adjustment. This outcome indicates that, within the spectral interval used for retrieval and at the spatial resolution of MAJIS, photometric effects have little influence on the thermal inversion. The Moon's global shape can be approximated by a smooth ellipsoid at the few-hundred-meter pixel scale of MAJIS, so the primary

driver of variability in the thermal signal is the distribution of insolation rather than geometric scattering. The Lommel–Seeliger law, while traditionally adopted for the diffuse scattering properties of the lunar regolith and effective at moderate phase angles where single scattering dominates, becomes less appropriate at the extreme phase angles encountered during the JUICE flyby. In this geometry, multiple scattering, surface roughness, and shadowing effects grow increasingly important, and a single-




parameter correction cannot capture such complexity. The absence of significant differences between corrected and uncorrected retrievals is therefore unsurprising and further supports the robustness of the Bayesian thermal inversion under the observing conditions of the JUICE flyby.

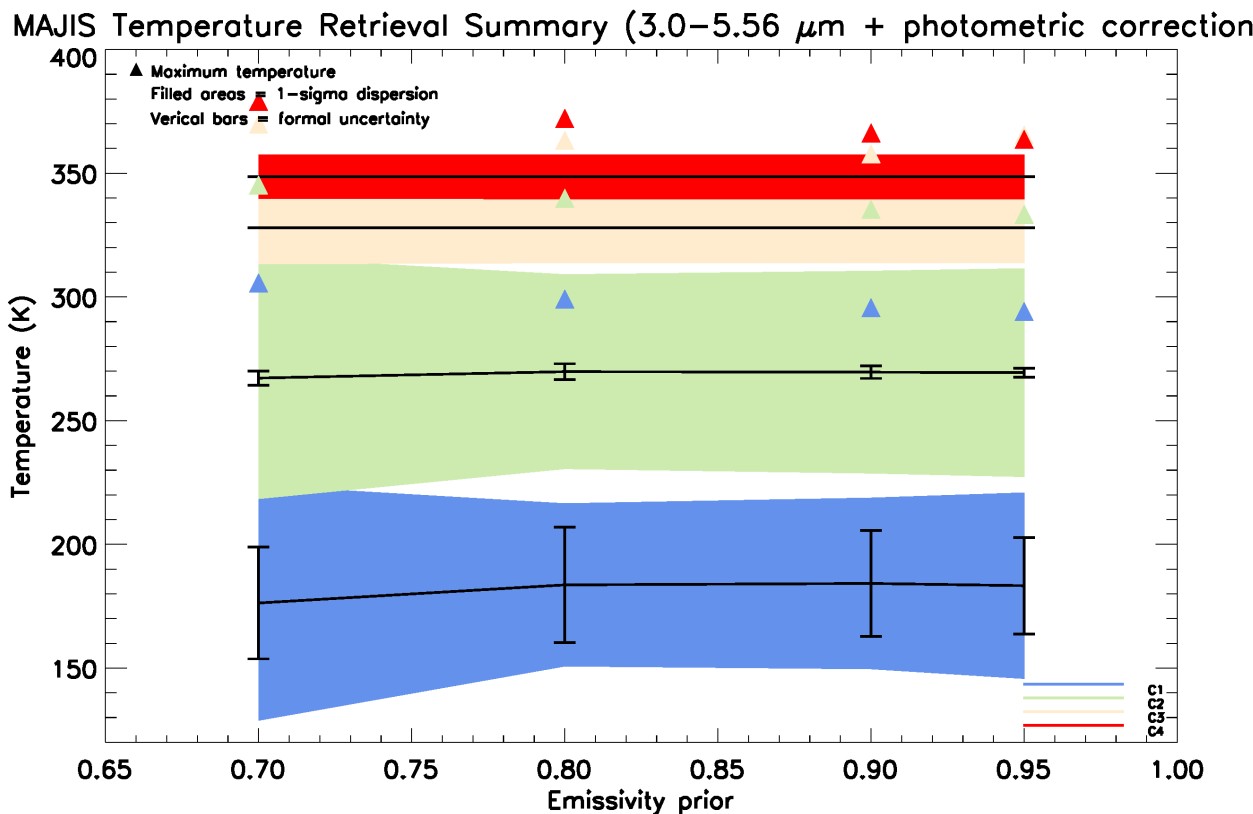

**Figure 6. Summary of MAJIS temperature retrievals as a function of emissivity prior (3.0–5.56 µm range, Lommel–Seeliger**
**correction applied).** The coloured bands indicate the mean surface temperatures (solid lines) and associated ±1σ dispersion for the four MAJIS observations (C1–C4), derived from Bayesian inversion including the Lommel–Seeliger photometric correction. Vertical black bars show the mean formal uncertainties, and triangles mark the maximum retrieved temperature values. Compared to the uncorrected case, the photometric correction yields slightly smoother temperature trends and marginally reduced variability, especially across high incidence geometries.


For the subsequent intercomparison with the other methods (Section 2.5), we adopt the Bayesian retrieval parameters that provide the most physically plausible and stable results: the 4.5–5.56 µm spectral range, an emissivity prior of $\varepsilon_0 = 0.70 \pm 0.15$, and no photometric correction.






## 2.2 Empirical thermal correction

Li and Milliken (2016) developed an empirical thermal correction model tailored for spectra acquired by the Moon Mineralogy Mapper (M³), which operated over the spectral range 0.43–3.0 µm (Pieters et al., 2009). They identified that at wavelengths beyond ~2 µm, thermal emission from the lunar surface contributes significantly to the measured radiance, particularly on the
dayside. This thermal component can obscure or distort diagnostic absorption features associated with minerals and hydration, rendering accurate compositional analysis challenging.

To address this, they constructed a model grounded in laboratory measurements of lunar soils and glasses, leveraging the strong empirical correlation between reflectance at 1.55 µm (unaffected by thermal emission) and at 2.54 µm (where thermal emission may be substantial, depending on the surface temperature). This relationship follows a power law, enabling the
prediction of the "true" reflectance at 2.54 µm based on the value at 1.55 µm. Any excess radiance at 2.54 µm is attributed to thermal emission, which can then be quantified and removed. Crucially, this approach allows for simultaneous estimation of the surface temperature, without requiring independent thermal measurements from instruments like Diviner, provided the observed materials match common lunar regolith compositions.

This method is particularly useful when applied to MAJIS data acquired in its visible and near-infrared (VISNIR) channel
covering the spectral range 0.49–2.36 µm. While this does not extend as far into the thermal regime as M³ (and the MAJIS IR channel itself), thermal contributions can still become non-negligible near the upper end of this spectral window—especially for observations near local noon or in equatorial regions. Applying the Li and Milliken (2016) model to MAJIS data enables the detection and removal of this thermal component, improving the fidelity of the retrieved reflectance spectra. In doing so, it not only enhances mineralogical interpretations in the near-IR but also yields estimates of surface temperature derived
directly from the observed radiance, assuming that a portion of the MAJIS spectrum captures the onset of thermal emission. This makes the approach particularly attractive for MAJIS lunar observations where Diviner-like thermal constraints may be unavailable or spatially mismatched, while allowing a direct comparison with the Bayesian method.

By grounding the correction in the inherent spectral behaviour of lunar soils and leveraging the statistical predictability of thermally unaffected wavelengths, this model provides a robust and computationally efficient tool for disentangling thermal
and reflective components in hyperspectral datasets such as those from MAJIS.

Formally, the method proposed by Li and Milliken (2016) relies on two main equations. The first is Kirchhoff's law of radiative equilibrium:

$$I = 1/\pi \, F_{Sun} \, R + I_{BB}(T)(1-R) \tag{1}$$






where $I$ is the total radiance emitted by the surface, $F_{Sun}$ the solar irradiance received by the surface, $I_{BB}$ the blackbody radiance at temperature $T$ associated with the surface thermal emission, and $R = 1-\varepsilon$ the surface reflectance, which is related to the emissivity $\varepsilon$ under radiative equilibrium. The second equation is an empirical power-law relationship between the reflectances at 1.55 µm and 2.54 µm, measured in the laboratory on lunar soil and glass samples under the geometry $i = 30°$, $e = 0°$:


$$R_{2.54} = 1.124 \times R_{1.55}^{0.8793} \tag{2}$$

Assuming that thermal emission is negligible at 1.55 µm, i.e., $I_{1.55} \approx 1/\pi \; F_{Sun} \, R_{1.55}$, the thermal emission of the surface—and therefore its temperature—can be directly derived from the radiances measured by MAJIS at 1.55 and 2.54 µm, together with
the incoming solar flux, without requiring additional assumptions about the surface emissivity. This makes the method computationally very efficient.

Nevertheless, over the range of lunar surface temperatures, the radiance excess at 2.54 µm can be small or even undetectable due to the radiometric uncertainty of MAJIS (see Langevin et al., this issue), which tends to scatter data points around the power-law curve (Figure 7). Consequently, all data points lying below the power-law relation correspond to surface
temperatures that cannot be reliably estimated—or are highly uncertain—using this empirical approach. This limitation is most pronounced near the terminator (e.g., in C2), where thermal emission is weak and radiometric uncertainties dominate.

To assess the accuracy of the surface temperature estimates, in Figure 7 we compare our results with those predicted by the two-layer thermal equilibrium model proposed by Vasavada et al. (1999), with updated parameters based on LRO/Diviner observations (Vasavada et al., 2012). Our surface temperature estimates show good agreement with the model for solar
incidence angles below 55° but diverge significantly at higher incidence angles. This discrepancy likely arises from unresolved topographic effects, which are not accounted for when data are georeferenced to a smooth ellipsoid. Sub-pixel slopes and shadowing can dominate the measured radiance and produce local deviations from the expected thermal behavior. In the MAJIS observations, this incidence range corresponds to the highlands observed at the end of C3 and the beginning of C4. In terms of accuracy, assuming a 10% uncertainty in the radiometric flux measured by MAJIS, the corresponding uncertainties
in surface temperature are estimated at ±15 K for C2, ±7 K for C3, and ±5 K for C4.

The Li and Milliken (2016) method also allows for an a posteriori determination of emissivity at 5.5 µm by inverting Kirchhoff's law, under the assumption that reflected sunlight at this wavelength is negligible, i.e., $I_{5.5} \approx \varepsilon_{5.5} \, I_{BB}(T)$. This capability will be further exploited in subsection 2.5.2, where emissivity retrievals obtained with different approaches are compared in detail.






**Figure 7. Empirical thermal correction results.** Lunar surface temperature estimates on the MAJIS' C2–4 observations. (**a**): map of the temperature along MAJIS' tracks. Background image: Moon LRO LROC WAC Global Morphology Mosaic 100m, June 2013 (Speyerer et al., 2011; Wagner et al., 2015). (**b**): reflectance at 2.54 µm as a function of the reflectance at 1.55 µm illustrating the power law relation established by Li and Milliken (2016). Points above the black curve correspond to a "reflectance excess" due to thermal emission. (**c**): derived surface temperature and its median (black line) as a function of solar incidence. The red curve is the Vasavada et al. (1999, 2012) temperature model during the lunar day.

## 2.3 Roughness-informed thermal model

An additional approach builds upon the roughness-informed model introduced by Wohlfarth et al. (2023), originally developed to improve our understanding of the lunar diurnal water cycle based on M³ observations. In this work, the framework is





specifically adapted to MAJIS data with the goal of retrieving surface temperature and emissivity, rather than performing a full thermophysical inversion.

Unlike empirical approaches, this model explicitly accounts for local topography through a fractal surface formulation that includes self-heating, self-scattering, sub-pixel shadowing, and bolometric albedo. In this fractal representation, the surface is depicted as a series of discrete square plates, called "*facets*." The model solves the energy balance for each facet $m$, where the total flux $F_m$ is the sum of three components: direct solar irradiation, scattered sunlight, and thermal self-heating from other facets. In our case, we divide each MAJIS pixel into a 65×65 sub-pixel grid, where the thermal balance is computed individually. The resolution of the sub-pixel facets provides a good compromise between capturing sub-pixel temperatures and computational tractability.

The parameters used to generate the synthetic fractal surfaces are constrained by macroscopic roughness values derived from LOLA Digital Roughness Map (LRO/LOLA LDRM_32 data) (Smith et al., 2010, 2017). To match the spatial resolution of MAJIS cubes, they have been interpolated using a Gaussian filter (Figure 8). This procedure yields slope distributions that are statistically consistent with pixel-scale roughness, avoiding implementations based on arbitrary assumptions or retrieved values of roughness. This approach reduces the risk of degeneracies among free parameters, which in this case are limited to pixel emissivity and temperature. In addition, effective incidence angles are computed by combining the observing geometry with local terrain slopes from LOLA Digital Slope Map (LRO/LDSM_16 data) (Smith et al., 2010, 2017), thereby better reproducing the actual physical illumination conditions (Figure 9).








**Figure 8. LOLA-derived roughness maps.** Estimated large-scale roughness derived from LDRM_32 LRO/LOLA data, interpolated with a Gaussian filter to match the spatial resolution of the four MAJIS observations: (**a**) C1 and C2; (**b**) C3 and C4. The estimated pixel-scale roughness values have been employed as a priori surface parameters to constrain the temperature estimation within the roughness-informed model approach. Background image: Moon LRO LROC WAC Global Morphology Mosaic 100m, June 2013 (Speyerer et al.,

2011; Wagner et al., 2015).



**Figure 9. Effective solar incidence angles.** Effective incidence angles derived combining the observational geometries with the local terrain slopes from LDSM_16 LRO/LOLA data, convolved to match the spatial resolution of the MAJIS observations: (**a**) C1 and C2; (**b**) C3 and C4. The variability associated with local topography is most evident in C1 and C2, where the altimetric variations measured by LRO/LOLA significantly alter solar incidence across rugged terrains. Background image: Moon LRO LROC WAC Global Morphology Mosaic 100m, June 2013 (Speyerer et al., 2011; Wagner et al., 2015).

Once the sub-pixel synthetic fractal roughness surface is generated for each pixel and the DEM-derived, effective solar incidence and emission angles, the thermal contribution of the *m*-th sub-pixel facet is given by the energy balance equation:

$$F_m = F_m^{sun} + F_m^{sca} + F_m^{rad} \tag{3}$$





where $F_m^{sun}$ is the flux of directly absorbed solar radiation, $F_m^{sca}$ is the contribution of scattered sunlight from neighbouring
sub-pixel facets and $F_m^{rad}$ is the additional flux due to thermal self-heating. The thermal emission related to direct solar
irradiation of the $m$-th sub-pixel facet is modelled as:

$$F_m^{sun} = (1 - A_{dh})J_m \tag{4}$$

where $A_{dh}$ is the directional–hemispherical albedo (Shkuratov et al., 2011) and $J_m$ is the incident solar irradiance on the $m$-th
sub-pixel facet. The terms for scattering from neighbouring sub-pixel facets ($F_m^{sca}$) and thermal self-heating ($F_m^{rad}$) are
respectively:

$$F_m^{sca} = (1 - A_{dh})A_{dh} \sum_{j \neq m}^{65x65} f_{m,j} J_j + (1 - A_{dh})A_{dh}^2 \sum_{j \neq m}^{65x65} f_{m,j} \sum_{k \neq j}^{65x65} f_{j,k} J_k + \ldots \tag{5}$$


$$F_m^{rad} = (1 - A_{dh,th}) \sum_{j \neq m}^{65x65} f_{m,j} F_j + (1 - A_{dh,th})A_{dh,th} \sum_{j \neq m}^{65x65} f_{m,j} \sum_{k \neq j}^{65x65} f_{j,k} F_k + \ldots \tag{6}$$

with $f_{i,j}$ are the geometric view factors (Wohlfarth et al., 2023) and $A_{dh,th}$ is the hemispherical albedo for thermal radiation,
which is linked to the effective emissivity $\varepsilon$ by Kirchhoff's law (Spencer, 1990). The reported equations are resolved in a
matrix-vector notation to account for higher order terms for $F_m^{sca}$ and $F_m^{rad}$, following the pipeline described by Wohlfarth et
al. (2023). Subsequently, the Stefan–Boltzmann law relates the radiant flux of the $m$-th sub-pixel facet $F_m$ to its equilibrium
temperature, which can be retrieved jointly with the effective emissivity $\varepsilon$:

$$T_m = \left(\frac{F_m}{\sigma \varepsilon}\right)^{\frac{1}{4}} \tag{7}$$


Once the temperatures of the sub-pixel facets are estimated, the thermal emission component linked to the $n$-th pixel is
defined as:

$$X_n(\lambda) = \frac{\sum_m^{65x65} P_m(\lambda) v_m \cos e_m}{\sum_m^{65x65} v_m \cos e_m} \tag{8}$$


where $P_m(\lambda)$ denotes the Planckian thermal emission of the $m$-th sub-pixel facet, computed from its estimated equilibrium
temperature; $v_m$ is a line-of-sight visibility factor to the sensor and $e_m$ is the emission angle of the $m$-th sub-pixel facet.
The retrieval is performed in the 5.0–5.5 μm spectral range, which is dominated by thermal emission and largely unaffected
by detector saturation. Within this range, the emissivity is assumed to be spectrally constant. An initial value of $\varepsilon = 0.95$ is
adopted, consistent with previous studies on lunar regolith (Bandfield et al., 2015; Bandfield et al., 2018) and subsequently



allowed to vary together with surface temperature. Both parameters are jointly optimized by minimizing the root mean square error (RMSE) between the modelled thermal emission $X_n(\lambda)$ and the observed MAJIS spectra.

The simulations are not performed on a pixel-by-pixel basis because the computational cost would be prohibitive, even under aggressive parallelization. Instead, we construct a three-dimensional grid whose axes correspond to the effective incidence
angle $i$, the emission angle $e$, and the macroscopic roughness $\rho_{rough}$. The internal spacing of the grid is set to steps of ($\Delta i$=0.5°; $\Delta e$=0.5°; $\Delta \rho_{rough}$=0.02). This discretized grid allows temperatures to be evaluated only for the ($i$, $e$, $\rho_{rough}$) combinations that occur within the hyperspectral cube. This approach avoids repeated computations for near-duplicate configurations that would yield comparable results and substantially reduces the overall computational cost. Because the phase angle for all four hyperspectral cubes is ~90° with minimal variability (on the maximum order of ±2° with respect to its average
value), it is not included as a fourth grid dimension. Instead, the mean phase angle for each ($i$,$e$,$\rho_{rough}$) combination is adopted as a fixed value in the simulations.

Temperatures retrieved from each MAJIS hyperspectral cube (Figure 10) exhibit a clear spatial correlation with both macroscopic surface roughness (Figure 8) and effective illumination conditions (Figure 9). The highest temperature value (385 K) is observed in the northwestern sector of C4, consistent with the local incidence angles and in agreement with the
location of maximum temperature identified by the other two methods. Conversely, the lowest temperature values (<170 K) occur in shadowed regions of C1 and C2, corresponding to pixels characterized by effective incidence angles exceeding 90° (Figure 10).







**Figure 10. Temperatures from roughness-informed model.** Temperatures retrieved using the roughness-informed thermal model: (**a**) C1 and C2; (**b**) C3 and C4. Spatial variations in the temperatures are primarily controlled by both surface roughness (Figure 8) and effective illumination conditions (Figure 9). The maximum temperature value (385 K) is observed in the northwestern portion of C4, whereas the lowest values (<170 K) occur in C1 and C2, within areas strongly affected by local topographic shading, corresponding to non-illuminated surfaces ($i > 90°$), as highlighted in Figure 9. Background image: Moon LRO LROC WAC Global Morphology Mosaic 100m, June 2013 (Speyerer et al., 2011; Wagner et al., 2015).

Accounting for macroscopic roughness introduces variability in the retrieved temperatures, even under identical illumination geometries. Here, the analysis focuses on C3 and C4, which provide the most favourable illumination conditions. For each observation, polynomial functions are fitted to the temperatures as a function of the effective incidence angle (Figure 11), and the resulting best-fit curves are grouped into subcategories representative of distinct macroscopic roughness classes. In general, temperatures increase with roughness, with the largest variability occurring near an incidence angle of ~55°. At higher





incidence angles, the growing role of local topography (Figure 11a) is consistent with the deviations observed in the empirical approach relative to the Vasavada model (Figure 7), highlighting that neglecting topographic effects—and thus thermal beaming (Spencer, 1990; Rozitis and Green, 2011)—introduces systematic biases in temperature retrievals. This trend should,

however, be interpreted with caution, as the roughness–temperature relationship is also likely modulated by factors not explicitly considered in this analysis, including true sub-pixel topography, surface composition, and wavelength-dependent emissivity.

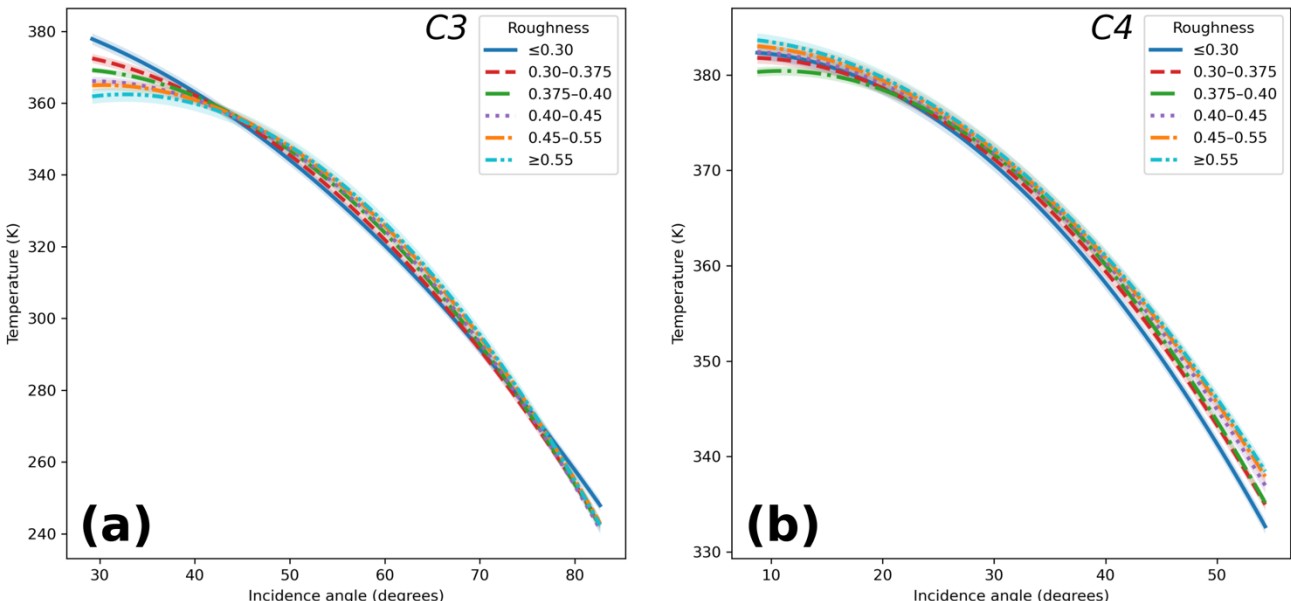

**Figure 11. Temperature as function of geometry (thermal beaming effect).** Polynomial fits of the retrieved temperatures, with $1\sigma$ uncertainties represented as shaded areas, are shown as a function of illumination geometry and macroscopic roughness for (**a**) C3 and (**b**) C4. In both cases, the thermal beaming effect reaches its maximum influence on surface temperature at an incidence angle of ~55°.

## 2.4 Cross-wavelength comparison with LRO/Diviner

To place the MAJIS thermal observations into a broader lunar context, we compared our results with co-located datasets from the Diviner Lunar Radiometer Experiment aboard LRO (Williams et al., 2017). Diviner provides radiance and brightness temperature for channels 3–6 (7.8–25 µm), along with derived emissivity maps and the Christiansen Feature (CF) wavelength. For this work, we gathered Diviner radiances for channels 3–6 acquired under sub-solar longitudes between –26° and +18°. Following Williams et al. (2018), the field of view of each measurement was modelled using a Monte-Carlo distribution of 50

points to compute its effective footprint.



Radiances were then adjusted to the solar distance corresponding to the JUICE flyby (1.014 AU) under the assumption of radiative equilibrium, and subsequently gridded at 128 pixels per degree. Because Diviner does not provide full spatial coverage at the exact solar geometry of the MAJIS observations (sub-solar longitude –4.1°, sub-solar latitude 1.0°), we estimated gap-free radiances by linearly interpolating the observations bracketing each pixel in sub-solar longitude. This procedure reproduces the large-scale thermal structure well, though orbit-to-orbit striping remains visible in places (Figure 12). CF position and peak brightness temperature were determined by parabolic fitting of channels 3, 4 and 5 (Williams et al., 2018), and channel emissivities were evaluated relative to the CF peak temperature. This approach highlights compositional and textural contrasts between mare and highlands, although emissivity estimates at large incidence angles ($i > 60°$) are strongly affected by photometric roughness effects, particularly in channel 6.








**Figure 12. Diviner brightness temperature and emissivity.** (**a**) Diviner peak brightness temperature (channels 3–6, 7.8–25 µm) over the MAJIS ground track, gridded at 128 px/degree (~240 m/px at the equator) and interpolated across adjacent sub-solar longitudes to match the illumination of the JUICE flyby. The map captures the large-scale thermal contrast between Mare Tranquillitatis and the surrounding highlands, with minor orbit-to-orbit striping. (**b**) Emissivity at 7.8 µm (Diviner channel 3), computed relative to the Christiansen Feature using the standard CF-fitting procedure. This wavelength is sensitive to compositional and textural differences. Background image: Moon LRO LROC WAC Global Morphology Mosaic 100m, June 2013 (Speyerer et al., 2011; Wagner et al., 2015).



## 2.5 Comparison of the three methods

### 2.5.1 Temperature

The comparison between three independent approaches for estimating lunar surface temperature from MAJIS data reveals systematic differences that depend on solar incidence angle and the radiometric quality of the observations. Figure 13 summarizes the results of (1) Bayesian inversion (Tosi et al., 2014), (2) an empirical correction based on laboratory-derived relationships (Li and Milliken, 2016), and (3) a roughness-informed thermal model (Wohlfarth et al., 2023). The latter approach requires discretizing the observational parameter space into sub-categories of incidence, emission, and roughness to ensure computational feasibility, which leads to the gridded appearance visible in Figure 13b.

The comparison among the three retrieval approaches highlights systematic differences that depend strongly on solar incidence angle and illumination conditions. The empirical correction following Li and Milliken (2016) increasingly diverges from the Vasavada thermal model at incidence angles beyond ~55°. This behaviour likely reflects the fact that the correction law was derived from laboratory spectra acquired under fixed geometry ($i = 30°$, $e = 0°$), conditions that differ markedly from the high phase angle (~90°) and variable incidence sampled during the JUICE flyby. By contrast, the Bayesian inversion shows greater robustness at high incidence and low-SNR data, producing smoother and more conservative temperature estimates that remain relatively close to the Vasavada curve. However, a notable deviation between Methods 1 and 2 and the Vasavada model is observed in the longitude range 30–40° corresponding to highland terrains observed in C4 under incidence angles in the range 35°–45°, where surface roughness and anisothermality introduce systematic biases. The roughness-informed Model 3 (Wohlfarth et al., 2023) generally yields the closest agreement with both Diviner and Vasavada temperatures under well illuminated conditions ($i < 60°$). Its explicit treatment of radiative balance and surface roughness makes it less sensitive to viewing geometry than the empirical method, while avoiding the smoothing bias of the Bayesian inversion. Nonetheless, in shadowed or high incidence regions, even this approach remains challenged by anisothermality and unresolved sub-pixel topography, which introduce localized deviations.

Taken together, the three methods illustrate complementary strengths. The empirical approach is most accurate when illumination conditions resemble those of the laboratory calibration but is prone to error at high incidence. The Bayesian inversion is robust under challenging radiometric conditions, although it may underestimate surface temperatures at intermediate incidence angles, reflecting the strong regularization imposed by the a priori covariance, which can suppress local spectral gradients. The roughness-informed thermal model achieves the best overall consistency with Diviner and Vasavada, yet still inherits limitations from geometric roughness and phase-angle extremes. The systematic differences among the three approaches underscore the importance of cross-validation and demonstrate that combining them yields the most reliable interpretation of MAJIS-derived lunar surface temperatures across diverse observational geometries and terrains.





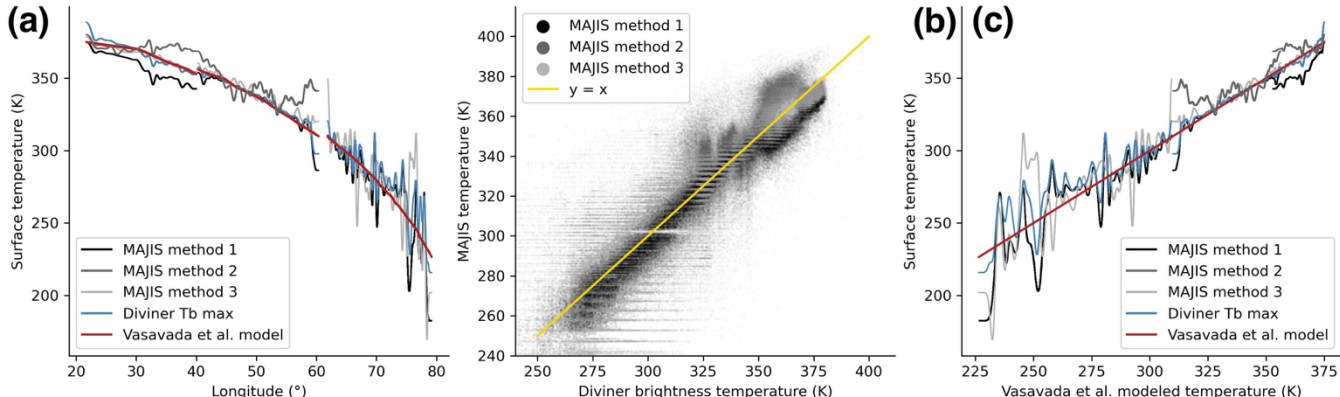

**Figure 13. Comparison of temperature retrieval methods.** Surface temperature estimates from MAJIS infrared data obtained with three independent approaches: (1) Bayesian inversion following Tosi et al. (2014) (black), (2) empirical correction following Li and Milliken (2016) (gray), and (3) a roughness-informed thermal model following Wohlfarth et al. (2023) (light gray). (**a**): retrieved temperatures along the JUICE ground track as a function of longitude, compared with Diviner maximum brightness temperatures (blue) and the thermophysical model of Vasavada et al. (1999) (red). (**b**): pixel-by-pixel comparison between MAJIS-derived temperatures and Diviner brightness temperatures, with the one-to-one line in yellow. The discretized appearance of Method 3 in this panel results from its computational design: pixels are grouped into sub-categories on a 3D grid of incidence, emission, and roughness parameters, while assuming a constant average phase angle, to ensure tractable run times. (**c**): comparison between MAJIS- and Diviner-derived temperatures, and the Vasavada et al. (1999) model. All methods converge within a few kelvins under well illuminated conditions, though larger deviations occur near the terminator.

### 2.5.2 Emissivity

Figure 14 presents a comparative analysis of surface emissivity derived from the MAJIS lunar observation C4, which was acquired under the most favourable illumination conditions. This dataset offers the best opportunity to assess emissivity retrievals, given the temperature uncertainties discussed in subsection 2.5.1. Three different approaches are considered: (1) the Bayesian inversion developed by Tosi et al. (2014), constrained by radiative transfer modelling; (2) the empirical thermal correction method derived from Li and Milliken (2016); and (3) a roughness-informed thermal model following Wohlfarth et al. (2023), adapted to the geometry of the JUICE flyby.





**Figure 14. Comparison of emissivity retrieval methods.** Emissivity estimates for the C4 observation obtained with the Bayesian approach or Method 1 (**a**), the Li and Milliken method or Method 2 (**b**), and the roughness-informed thermal model or Method 3 (**c**), respectively. Panel (**d**) shows a comparison between the emissivity estimated with Method 1 (blue dots), Method 2 (red dots) and Method 3 (green dots) as a function of longitude. Background image: Moon LRO LROC WAC Global Morphology Mosaic 100m, June 2013 (Speyerer et al., 2011; Wagner et al., 2015).

Figure 14 compares the emissivity retrievals from the three methods in both map view and as longitudinal profiles. Figures 14a–c present the spatial distribution of emissivity at 5.5 μm derived with the Bayesian inversion (Method 1), the empirical correction (Method 2), and the roughness-informed thermal model (Method 3), respectively. In Method 1, full spectral emissivity profiles between 4.5 and 5.5 μm are retrieved using a prior of 0.70 ± 0.15, and only the scalar value at 5.5 μm is mapped. Method 2 computes emissivity at the same wavelength directly from Kirchhoff's law, assuming negligible reflected





sunlight, whereas Method 3 retrieves emissivity over 5.0–5.5 µm by explicitly including roughness and illumination geometry
in the forward radiative transfer. Panel 14d summarizes these results as a function of longitude, allowing direct comparison of
terrain-dependent variability among the three approaches.

All methods detect the same first-order geophysical signal: Mare Tranquillitatis exhibits higher emissivity than the surrounding
highlands. However, the magnitude and variability of emissivity differ substantially across methods. The Bayesian inversion
(blue dots in Figure 14d) yields values tightly clustered around $\varepsilon \approx 0.85$ in the mare and 0.75–0.85 in the highlands with
minimal scatter, reflecting the smoothness constraint imposed by the prior covariance. The empirical method (red dots)
produces significantly lower values (0.60–0.70 in the mare and 0.40–0.60 in the highlands) and shows enhanced pixel-to-pixel
noise. Method 3 (green dots) returns similar values—$\varepsilon \approx 0.60$–0.70 in the mare and $\varepsilon \approx 0.50$–0.80 in the highlands—with
greater variability in rough or cratered areas. This variability is consistent with the inherently directional nature of emissivity
and the influence of surface slopes on thermal emission (e.g., Rozitis and Green, 2011).

Although Methods 2 and 3 both assume spectrally neutral emissivity over their retrieval interval, their behaviour in highland
regions differs markedly. Method 3 systematically yields higher emissivity than Method 2, falling between the Bayesian and
empirical estimates. This indicates that the underestimation by Method 2 is not solely due to the grey-emissivity assumption,
but also to its neglect of roughness (and associated beaming effects), its reliance on laboratory calibration relationships
obtained at fixed geometry, and the propagation of uncertainties from the multiple wavelengths used in the correction. By
explicitly accounting for geometry and roughness, Method 3 mitigates several of these limitations, even within a narrower
spectral range.

The spatial maps further illustrate these differences in retrieval fidelity. The Bayesian-derived map (Figure 14a) shows a
smooth and coherent emissivity pattern that closely reflects the large-scale thermophysical contrast between mare and
highlands. The empirical map (Figure 14b) displays stronger noise and variability, particularly over highland terrains, while
the roughness-informed model (Figure 14c) preserves the large-scale trend and captures localized fluctuations in rough areas
and along crater rims, consistent with its explicit sensitivity to surface geometry. Despite their methodological differences, all
three approaches reproduce the same first-order emissivity dichotomy between mare and highlands.

This contrast aligns with known compositional and textural differences. Feldspathic highland regoliths are more porous and
finely comminuted, enhancing multiple scattering and lowering effective emissivity, whereas denser, smoother basaltic mare
surfaces exhibit higher emissivity (e.g., Donaldson Hanna et al., 2012). This interpretation is consistent with full-disk
observations from NOAA/HIRS (3.75–15 µm) analysed by Müller et al. (2021), who reported systematically lower emissivity
over highlands—particularly beyond 5 µm—and attributed this to variations in composition, grain size, and roughness. The
agreement between MAJIS-derived emissivity and independent long-wavelength datasets provides strong validation of the
observed mare–highland dichotomy and underscores the value of combining multiple retrieval strategies to isolate
instrumental, geometric, and geophysical effects.

When compared with Diviner, a clear wavelength dependence emerges. Diviner channel 3 radiances and emissivity (7.8 µm)
show the opposite trend—highlands appearing more emissive than maria—consistent with the analysis of Ren et al. (2021)





and reflecting the well-established inversion across the Christiansen Feature. This behaviour is fully supported by laboratory spectra of lunar soils and analogues, which show that feldspathic materials become relatively more emissive beyond ~7–8 µm,

whereas basaltic surfaces dominate at shorter wavelengths. Retrievals from Chandrayaan-2/IIRS in the 3–5 µm range (Ojha et al., 2024) similarly report emissivity values between 0.60 and 0.80 with pronounced spatial variability at shorter wavelengths, reinforcing the wavelength-dependent behaviour observed by MAJIS.

Together, these datasets indicate that highlands exhibit lower emissivity than maria up to ~5.5 µm—the upper limit of MAJIS sensitivity—while beyond this wavelength the trend reverses. This spectral crossover, governed by the vibrational properties

of silicate minerals and by particle-scale roughness, reconciles the MAJIS and Diviner observations and highlights their complementarity.

The CF positions derived from Diviner additionally provide an independent compositional indicator that complements MAJIS hyperspectral retrievals at shorter wavelengths, reinforcing the distinction between mare basalts and feldspathic highlands. The comparison thus demonstrates the synergy between the two instruments: Diviner offers long-wavelength, global coverage

with excellent diurnal sampling, while MAJIS provides high-resolution hyperspectral measurements in the 3.0–5.56 µm range, bridging the gap between near- and mid-infrared observations. Taken together, these datasets underscore the wavelength-dependent nature of lunar emissivity and the importance of multi-instrument analyses for disentangling the combined effects of composition, texture, roughness and thermal environment on airless-body surfaces.




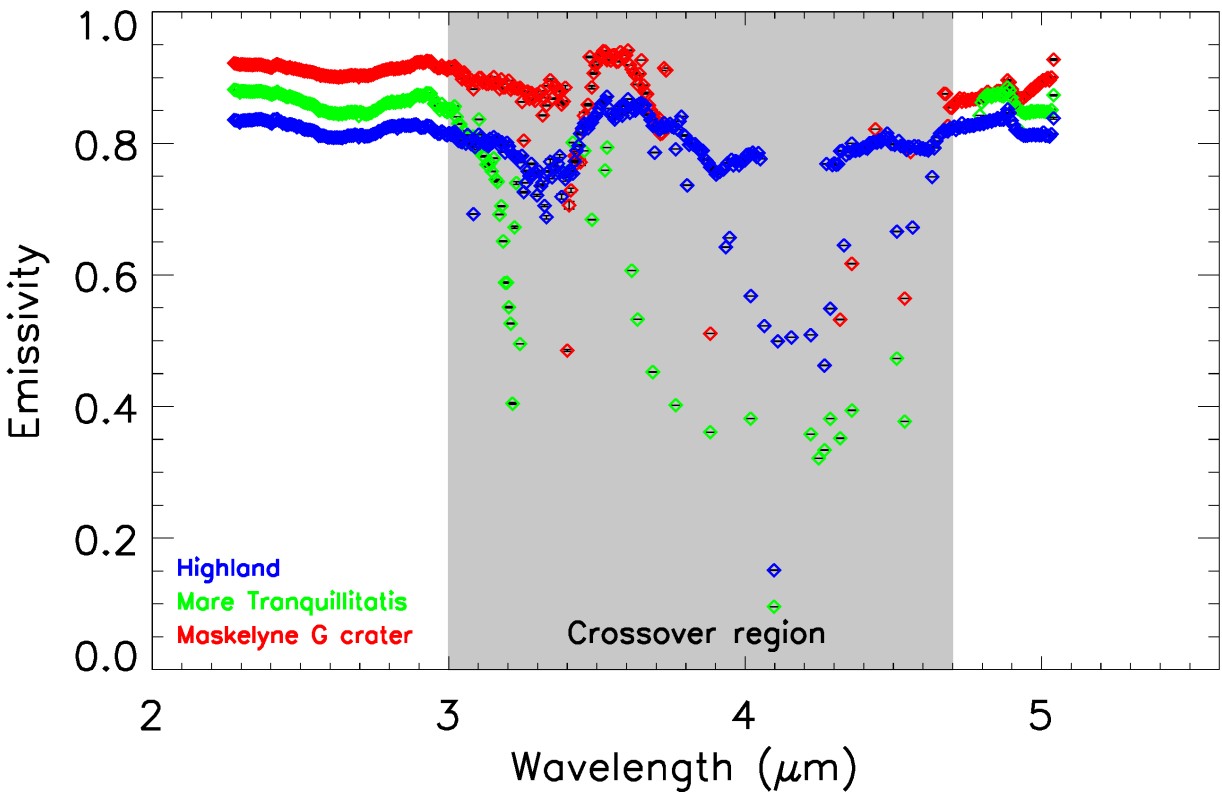

**Figure 15. Spectral emissivity of different terrains.** Spectral emissivity profiles retrieved from three lunar surface units using the Bayesian inversion applied to MAJIS data C4. The thermal retrieval is performed only over the 3.0–5.56 µm range, where the signal is dominated by emitted radiance. The resulting emissivity values are then extended to the full 2.0–5.56 µm interval by combining the retrieved emissivities with the photometrically corrected reflectance ($\varepsilon = 1 - r$). Shown are average emissivity profiles for a highland region (blue), Mare Tranquillitatis (green), and the Maskelyne G crater (red). Highland terrain displays the lowest emissivity, mare surfaces slightly higher values, and the crater the highest, consistent with blockier and less weathered material. Error bars (formal uncertainties) are very small and appear as black markers. The grey box highlights the "crossover region" (3.0–4.7 µm) where reflected sunlight and thermal emission are comparable.

Building on this result, Figure 15 presents representative spectral emissivity profiles extracted from three distinct lunar terrains—highlands, Mare Tranquillitatis, and Maskelyne G, a 6-km fresh crater located within the mare. These profiles reveal clear, systematic differences that reflect underlying variations in composition, grain size, and thermophysical properties. The spectra were derived using the Bayesian inversion method applied to MAJIS data in the 3.0–5.5 µm range, under the assumption of an emissivity prior of 0.70 ± 0.15 and no photometric correction. Across the spectral interval, the highland spectrum shows the lowest emissivity values, typically between 0.78 and 0.85, consistent with the known properties of



feldspathic materials. Mare Tranquillitatis exhibits higher emissivity across the same wavelengths, indicative of denser, basaltic surfaces with reduced porosity and higher thermal inertia. The Maskelyne G crater spectrum displays the highest emissivity, close to or above 0.90 throughout the thermal domain, consistent with exposure of coarser, less-weathered or blocky

material exhibiting reduced multiple scattering and enhanced thermal emission (Salisbury et al., 1987; Mustard and Hays, 1997; Donaldson Hanna et al., 2012), and potentially enhanced by localized self-heating and thermal-infrared beaming within concave or rough crater interiors (Rozitis and Green, 2011; Davidsson and Rickman, 2014). Mineralogical analysis of the same MAJIS dataset indicates that this crater is compositionally distinct from the surrounding terrain, showing an enhanced abundance of pyroxene-rich material, particularly pigeonite (Zambon et al., this issue).

While the emissivity differences between terrains are geologically meaningful, the presence of apparent absorption-like minima—particularly between 3.0 and 4.7 μm—requires careful interpretation. These features are not associated with known silicate vibrational bands, which typically occur beyond 8 μm, outside the MAJIS spectral range. Instead, they likely result from challenges in retrieving emissivity in the transition region where reflected solar and thermally emitted radiation overlap. In this "crossover region," small errors in radiometric calibration, assumed surface reflectance, or modelling of solar

contribution can propagate into the retrieval, leading to artificial depressions in emissivity. The variability in shape and depth of these minima across different terrains further supports their non-mineralogical origin. Although future improvements in reflectance modelling may mitigate these effects, under the current assumptions such features should be regarded as retrieval artefacts rather than true diagnostic absorption bands.

These results confirm that spectral emissivity in the 3–5 μm region—though shortward of the classic Reststrahlen bands—

encodes valuable information about lunar surface properties. The clear separation between the emissivity profiles retrieved from the three terrains underscores the potential of MAJIS for identifying subtle differences in surface roughness, texture, and thermal behaviour, particularly when integrated with high-resolution geological context and cross-wavelength validation from instruments such as Diviner.

## 3 Conclusions

The MAJIS observations obtained during the JUICE lunar flyby provide a unique opportunity to assess the thermal and emissivity properties of the Moon at high spatial and spectral resolution. By applying three independent approaches—Bayesian inversion (Tosi et al., 2014), empirical correction following Li and Milliken (2016), and a roughness-informed thermal model following Wohlfarth et al. (2023)—we derived surface temperatures and emissivity values that can be cross-validated against one another and compared with both Diviner datasets and thermophysical models.

The comparison among methods highlights both convergence and divergence. All three techniques reproduce the expected temperature increase with solar illumination, with agreement within a few kelvins in the most favourable radiometric conditions (C3–C4). However, their behaviour diverges at higher incidence angles or in low-SNR scenes. The empirical approach, being tied to laboratory-derived correction laws, tends to overestimate surface temperatures under geometries far



from its calibration range, particularly at $i > 55°$, where reflected and thermal contributions overlap, and noise levels rise. The
Bayesian approach, by contrast, remains more stable under these conditions but shows a tendency to underestimate peak
temperatures at intermediate incidence angles, a likely manifestation of over-regularization. The roughness-informed thermal
model provides the closest match to Diviner brightness temperatures and to the Vasavada et al. (1999, 2012) thermophysical
model under well lit conditions, though it is not immune to deviations in shadowed or anisothermal regions. These results
indicate that each approach has domain-specific strengths and that their joint application enhances the robustness of the thermal
interpretation.

A key outcome of this study is the characterization of emissivity across different lunar terrains. Despite methodological
differences, all approaches converge on the finding that mare basalts exhibit higher emissivity than surrounding highlands.
This contrast is physically meaningful: feldspathic highland regolith is more porous and finely comminuted, which enhances
multiple scattering and reduces effective emissivity, while maria, dominated by denser basaltic materials, display higher
thermal inertia and higher emissivity. Fresh crater interiors in Mare Tranquillitatis, such as Maskelyne G, reveal even higher
emissivity, pointing to blockier and less-weathered material with reduced porosity and enhanced conductivity.

The comparison with Diviner further refines this picture. At the shorter side of the Christiansen Feature sampled by MAJIS
(3.0–5.5 µm), maria are more emissive than highlands, while at 7.8 µm (Diviner channel 3) the pattern reverses, with highlands
appearing more emissive than maria. Laboratory data confirm that such inversions are characteristic of silicate regoliths:
feldspathic materials exhibit relatively higher emissivity near the CF, while basalts dominate at shorter wavelengths. This
spectral dependence reconciles the apparent contradiction between MAJIS and Diviner and underscores the importance of
considering wavelength-dependent emissivity when interpreting lunar surface properties.

The roughness-informed thermal model adds a critical dimension to this analysis by retrieving roughness maps alongside
temperature. These maps show enhanced roughness in highland terrains and around crater rims, while mare surfaces appear
smoother on average. Such spatial patterns are consistent with the emissivity contrasts described above and with expectations
from lunar geology. Importantly, roughness is not merely a static property but a factor that drives anisothermality: rough
surfaces contain a distribution of slopes and facets that experience different solar illumination, producing sub-pixel temperature
variations. These anisothermal conditions explain, at least in part, the divergence observed between the Bayesian and empirical
methods in highland terrains observed under moderated solar illumination. The empirical method, which assumes a simplified
radiative balance, fails to account for this complexity and in some cases may tend to overestimate surface temperatures. On
the other hand, the Bayesian approach, with its strong regularization, may tend to suppress the thermal variability introduced
by anisothermality, leading to underestimation of temperatures. The roughness-informed thermal model, by explicitly
incorporating roughness into its energy-balance framework, mitigates these effects more effectively, though localized
deviations remain in the most extreme geometries.

The thermophysical patterns retrieved by MAJIS show a clear correspondence with the mineralogical variability independently
identified in the VNIR domain (Zambon et al., this issue). In Mare Tranquillitatis, areas exhibiting deep 1–2 µm pyroxene
absorptions—indicative of Fe- and Ca-rich basalts—display higher emissivity and warmer daytime temperatures, consistent





with denser and less weathered regolith. In contrast, the feldspathic highlands, spectrally dominated by plagioclase and characterized by strong space-weathering reddening, show lower emissivity and more heterogeneous thermal behaviour. Local

thermal enhancements near fresh craters similarly match VNIR detections of compositionally fresh, blocky pyroxene-rich materials. These correlations indicate that the 3–5.5 µm emissivity signal is strongly influenced by composition and regolith maturity, in addition to illumination geometry and surface roughness.

Taken together, the MAJIS observations demonstrate that the instrument retrieves surface temperatures with high precision while also capturing regolith properties that control emissivity and roughness. The emissivity contrasts between maria and

highlands, the spectral inversion across the Christiansen Feature, and the explicit roughness estimates collectively reveal a thermophysical diversity closely linked to composition, grain size, and texture. These results show how the mineralogical variability observed across the lunar surface translates into distinct thermophysical behaviours.

In this context, MAJIS provides a bridge between compositional and physical interpretations of the lunar regolith. By coupling hyperspectral temperature and emissivity modelling, this dataset allows us to explore how mineralogy, grain size and roughness

jointly shape the thermal environment of the lunar surface. The synergy between MAJIS and Diviner further demonstrates the value of combining datasets across different spectral ranges. Together, they provide a multi-wavelength view of the Moon that is essential for constraining regolith properties, improving thermophysical models, and guiding the interpretation of thermal emission from other airless bodies.

This end-to-end pipeline, validated by multi-mission heritage, delivers a unique hyperspectral thermal dataset of the Moon.

Acquired at a high phase angle and with broad spectral coverage, it provides a critical benchmark for thermal retrieval methods. The successful disentanglement of temperature and emissivity under these conditions establishes a robust framework for MAJIS's exploration of the Jovian system. This capability will be pivotal for probing the thermophysical properties of the most promising targets, such as Callisto, where subsolar temperatures are predicted to produce a detectable thermal signal within the MAJIS spectral range (Royer et al., 2025), thereby extending the instrument's role from compositional mapping to

direct thermophysical characterization.




## Appendix A


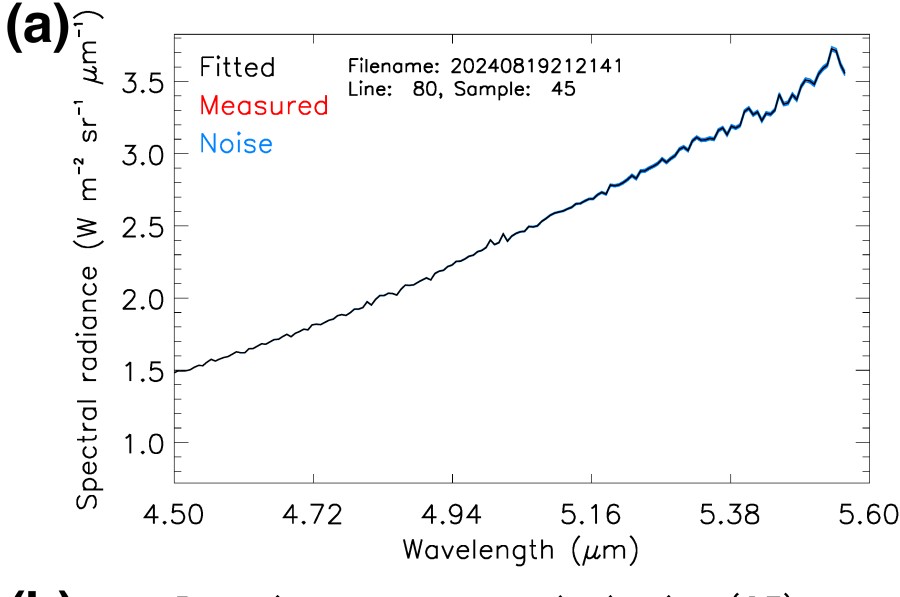

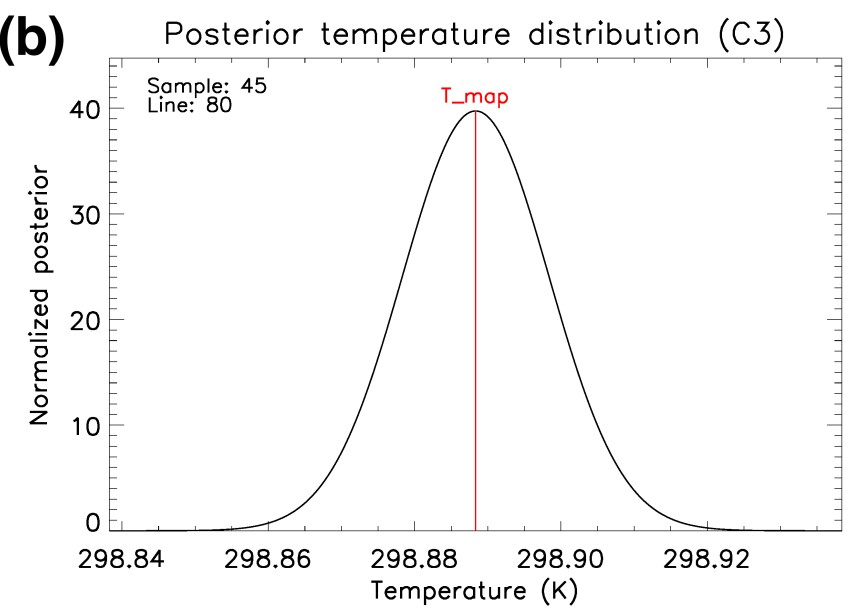

**Figure A1. Example of MAJIS thermal retrieval and posterior temperature distribution.** (a) Example of MAJIS radiance fit for pixel (sample = 45, line = 80) from the MAJIS C3 observation (file 20240819212141). The measured radiance (red, hidden beneath the model in black) is retrieved over the 4.5–5.5 μm range with an initial emissivity prior of 0.7. The NESR is shown in blue. The near-complete overlap of measured and modelled spectra highlights the accuracy of the thermal retrieval. (b) Posterior probability density function for the surface

temperature associated with the same pixel as panel (a). The black curve shows the temperature posterior, approximated as a Gaussian





centred at the maximum a posteriori estimate $T_{map}$, with width given by the formal uncertainty derived from the posterior covariance matrix. The red vertical line marks the retrieved temperature value ($T_{map} \approx 298.9$ K) for this pixel, which lies in a well illuminated region with very low formal uncertainty ($\sigma_t \approx 0.01$ K), producing a narrow and well-constrained posterior. This example illustrates the internal consistency

and stability of the thermal retrieval method applied to MAJIS NIR radiances.

**Table A1.** Summary of the temperature values obtained by applying the Bayesian approach to the four data acquired by MAJIS, using the 4.5–5.56 μm range and applying no photometric correction. The different columns specify the a priori hypotheses on the initial emissivity

($\varepsilon_0$) along with maximum associated standard deviation. For each MAJIS data, the following information is reported: mean temperature value, mean formal uncertainty associated with the data, 1-sigma dispersion of the data, and maximum temperature value. Units are kelvin.

| Observation | Parameter | $\varepsilon_0 = 0.7 \pm 0.15$ | $\varepsilon_0 = 0.8 \pm 0.20$ | $\varepsilon_0 = 0.9 \pm 0.10$ | $\varepsilon_0 = 0.95 \pm 0.05$ |
|---|---|---|---|---|---|
| C1 | T_mean | 175.7 | 184.0 | 184.3 | 183.3 |
| | $\sigma$_formal | 22.5 | 23.1 | 21.3 | 19.5 |
| | $\sigma$_data | 48.5 | 32.5 | 34.6 | 37.6 |
| | T_max | 305.6 | 299.1 | 295.6 | 294.1 |
| C2 | T_mean | 267.8 | 270.0 | 269.4 | 269.2 |
| | $\sigma$_formal | 3.0 | 3.2 | 2.5 | 1.8 |
| | $\sigma$_data | 49.0 | 39.2 | 40.8 | 41.7 |
| | T_max | 345.2 | 339.9 | 335.4 | 333.4 |
| C3 | T_mean | 333.4 | 329.2 | 325.7 | 324.2 |
| | $\sigma$_formal | 0.8 | 0.4 | 0.4 | 0.2 |
| | $\sigma$_data | 16.1 | 14.6 | 13.4 | 12.8 |
| | T_max | 369.8 | 363.3 | 357.8 | 365.4 |
| C4 | T_mean | 355.9 | 350.1 | 345.2 | 343.0 |
| | $\sigma$_formal | 1.4 | 1.5 | 0.7 | 0.3 |
| | $\sigma$_data | 9.6 | 9.1 | 8.7 | 8.5 |
| | T_max | 379.0 | 372.1 | 366.3 | 363.7 |

**Table A2.** Summary of the temperature values obtained by applying the Bayesian approach to the four data acquired by MAJIS, using the

3.0–5.56 μm range and applying no photometric correction. The different columns specify the a priori hypotheses on the initial emissivity ($\varepsilon_0$) along with maximum associated standard deviation. For each MAJIS data, the following information is reported: mean temperature value, mean formal uncertainty associated with the data, 1-sigma dispersion of the data, and maximum temperature value. Units are kelvin.

| Observation | Parameter | $\varepsilon_0 = 0.7 \pm 0.15$ | $\varepsilon_0 = 0.8 \pm 0.20$ | $\varepsilon_0 = 0.9 \pm 0.10$ | $\varepsilon_0 = 0.95 \pm 0.05$ |
|---|---|---|---|---|---|





| C1 | T_mean | 176.3 | 183.6 | 180.2 | 183.3 |
|----|--------|-------|-------|-------|-------|
|    | $\sigma$_formal | 22.6 | 23.3 | 19.6 | 19.5 |
|    | $\sigma$_data | 47.5 | 32.9 | 39.7 | 37.6 |
|    | T_max | 301.6 | 298.5 | 293.6 | 294.2 |
| C2 | T_mean | 267.2 | 269.8 | 269.6 | 269.4 |
|    | $\sigma$_formal | 2.9 | 3.2 | 2.5 | 1.8 |
|    | $\sigma$_data | 48.8 | 39.3 | 40.9 | 42.1 |
|    | T_max | 340.1 | 405.2 | 340.1 | 340.1 |
| C3 | T_mean | 328.0 | 328.0 | 328.0 | 328.0 |
|    | $\sigma$_formal | 0.0 | 0.0 | 0.0 | 0.0 |
|    | $\sigma$_data | 14.7 | 14.3 | 14.2 | 14.2 |
|    | T_max | 362.3 | 430.9 | 365.6 | 365.7 |
| C4 | T_mean | 348.6 | 348.6 | 348.6 | 348.6 |
|    | $\sigma$_formal | 0.0 | 0.0 | 0.0 | 0.0 |
|    | $\sigma$_data | 8.9 | 9.0 | 9.0 | 9.0 |
|    | T_max | 383.1 | 399.3 | 382.0 | 378.8 |

**Table A3.** Summary of the temperature values obtained by applying the Bayesian approach to the four data acquired by MAJIS, using the 3.0–5.56 μm range and applying the Lommel–Seeliger photometric correction. The different columns specify the a priori hypotheses on the initial emissivity ($\varepsilon_0$) along with maximum associated standard deviation. For each MAJIS data, the following information is reported: mean temperature value, mean formal uncertainty associated with the data, 1-sigma dispersion of the data, and maximum temperature value. Units are kelvin.

| Observation | Parameter | $\varepsilon_0 = 0.7 \pm 0.15$ | $\varepsilon_0 = 0.8 \pm 0.20$ | $\varepsilon_0 = 0.9 \pm 0.10$ | $\varepsilon_0 = 0.95 \pm 0.05$ |
|-------------|-----------|--------------------------------|--------------------------------|--------------------------------|---------------------------------|
| C1 | T_mean | 176.3 | 183.6 | 184.2 | 183.3 |
|    | $\sigma$_formal | 22.6 | 23.3 | 21.4 | 19.5 |
|    | $\sigma$_data | 47.5 | 32.9 | 34.6 | 37.6 |
|    | T_max | 301.6 | 298.5 | 295.6 | 294.2 |
| C2 | T_mean | 267.2 | 269.8 | 269.6 | 269.4 |
|    | $\sigma$_formal | 2.9 | 3.2 | 2.5 | 1.8 |
|    | $\sigma$_data | 48.8 | 39.3 | 40.9 | 42.1 |
|    | T_max | 340.1 | 405.2 | 340.1 | 340.1 |
| C3 | T_mean | 328.0 | 328.0 | 328.0 | 328.0 |



| | | | | | |
|---|---|---|---|---|---|
| | $\sigma$_formal | 0.0 | 0.0 | 0.0 | 0.0 |
| | $\sigma$_data | 14.7 | 14.3 | 14.2 | 14.2 |
| | T_max | 362.3 | 430.9 | 365.6 | 365.7 |
| **C4** | T_mean | 348.6 | 348.6 | 348.6 | 348.6 |
| | $\sigma$_formal | 0.0 | 0.0 | 0.0 | 0.0 |
| | $\sigma$_data | 8.9 | 9.0 | 9.0 | 9.0 |
| | T_max | 383.1 | 399.3 | 382.0 | 378.8 |


*Code availability*. The IDL and Python codes used to retrieve surface temperature and emissivity values are direct implementations of published methods. These scripts were developed independently by specific authors (FT, CR, FC) for internal research purposes, are not publicly released, and may be shared on a case-by-case basis upon justified request.


*Data availability*. The MAJIS data acquired during the JUICE Moon–Earth flyby in August 2024 are currently under the mission's cruise-phase proprietary period. These data will be made available through the ESA Planetary Science Archive following the first Cruise Archive Delivery, which is currently scheduled for six months after Earth Gravity Assist #3 in 2029. Diviner data are publicly accessible from the NASA Planetary Data System (PDS) Geosciences Node: https://pds-
geosciences.wustl.edu/missions/lro/diviner.htm

*Author contributions*. FT, CR and FC carried out data analysis. FT prepared the manuscript with major contributions from CR and FC. TMP and BTG provided maps derived from Diviner data. FP, AM and FZ contributed to the discussion of results. FP and GP are respectively the PI and Co-PI of the MAJIS instrument. All authors have read and approved the manuscript.


*Competing interests*. The authors declare that they have no conflict of interest.

*Acknowledgements*. The authors wish to thank ESA teams from SOC, MOC and ESTEC, as well as Airbus Defence and Space for their technical and operational support to the MAJIS project. JUICE is a mission under ESA leadership with contributions
from its Member States, NASA, JAXA and the Israel Space Agency. It is the first Large-class mission in ESA's Cosmic Vision programme.

*Financial support.* FT, GP, AM, CC and FZ acknowledge support from the Italian Space Agency (ASI), implementation agreement ASI–INAF n. 2023-6-HH.0. CR, FP, YL and CP acknowledge support from the Centre National d'Études Spatiales
(CNES), contract CNES–CNRS n° 180 117.



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

            issue.