# Peer review of "The JUICE 2024 close flyby of the Moon: Thermal assessment from MAJIS"

_EGUsphere, 2025_

## Referee Comment (RC1)

Review of the manuscript (ID: egusphere-2025-6150) titled "The JUICE 2024 close flyby of the Moon: Thermal assessment from MAJIS" by Tosi et al. submitted to the journal *Annales Geophysicae (ANGEO)* for publication.

The manuscript reports the results of the retrieval of Lunar surface temperature and emissivity from the observations of Moons and Jupiter Imaging Spectrometer (MAJIS) onboard European Space Agency's JUpiter ICy moons Explorer (JUICE). Four high spatial resolution hyperspectral images, referred to as C1, C2, C3, and C4 in the manuscript, of the lunar surface were acquired in the spectral range 0.49-5.56 $\mu$m from MAJIS during JUICE's lunar flyby in August 2024. Authors have retrieved lunar surface temperature and spectral emissivity from these images using three different approaches: Bayesian inversion (Tosi et al., 2014), empirical correction following Li and Milliken (2016), and a roughness-informed thermal model following Wohlfarth et al. (2023). The intercomparison of retrievals from the three approaches is performed against one another. The retrieved surface temperature is compared with both Diviner observations and with those obtained from the thermophysical model. The roughness-informed thermal model approach is reported to be the most effective, followed by Bayesian and empirical correction methods.

Overall, the manuscript is well written considering all aspects of the study. While comparing the results, the authors have also highlighted the limitations of each algorithm. However, I have a few concerns that need to be addressed before its acceptance for publication in ANGEO.

**Technical Comments:**

1) The retrievals from the Bayesian approach are reported to be dependent on the *a priori*. It would be more informative if the authors also reported the degree of dependence. Sensitivity analysis of the retrievals on *a priori* will add more value to the retrievals. Such a kind of diagnostics is available from the algorithm itself and straightaway tell how much observations (and *a priori*) are contributing to the retrievals.

2) Bayesian inversion is all about giving proper weights to the observations and *a priori*. There is no mention of the covariance matrices used in the retrieval. Although the authors have mentioned the impact of the covariance matrix at many places in the text. Please mention how the covariance matrices (both observation and *a priori*) are prepared. Whether full covariance matrices are used for both observations and *a priori*? If only diagonal matrices are used, then what is done to compensate for the missing covariances? Hyperspectral observations are usually correlated. In practice, only the diagonal observation error covariance matrix is used, and errors are inflated. For *a priori* covariance matrix, the emissivity may be assumed to be uncorrelated, but temperature and emissivity are highly correlated, so their correlations must be included. Ignoring covariances will result in suboptimal retrievals. Physically inconsistent results mentioned on page no. 13, line no. 272 can be better explained with the help of this information.

3) It is not clear whether topographic corrections are implemented in the Bayesian approach for retrieval. The sentence on On Page no. 6, line 184, "Topographic effects may be mitigated..." is creating doubt. It's not certain. Further, the sentence on Page no. 7., line no. 185, "However, while such corrections are critical...a few hundred meters." is creating more suspicion. On Page no. 13, line no. 298, "The Moon's global shape can be approximated by a smooth ellipsoid...rather than geometric scattering." emphasize that

topographic effects are not important at MAJIS observations' spatial resolution, but on Page no. 16, line no. 370, "This discrepancy likely arises from unresolved topographic effects...can dominate the measured radiance and produce local deviations from the expected thermal behavior." and Page no. 23, line no. 491, "At higher ...in temperature retrievals", the authors attribute the retrieval errors to local topography.

4) The retrievals from the three algorithms mentioned in the study provide the physical lunar surface temperature, but LRO Diviner provides bolometric temperature. Authors have compared the surface temperature retrievals against the Diviner peak brightness temperature (determined by parabolic fitting of channels 3, 4, and 5). It is still brightness temperature not physical surface temperature. Authors should add a justification for making such a comparison and should also mention this while discussing discrepencies in the results.

5) In the "Roughness-informed thermal model" used in this study, it is not clear from the text whether whole model (Reflectance + Emission) given by Wohlfarth et al. (2023) is implemented or only the emission model is used. Wohlfarth et al. (2023) used Hapke model for reflectance which involves various parameters. Moreover, single scattering albedo is used as a free parameter in reflectance model. If whole model is used then the results are self explanatory.

6) Figure A1. in Appendix A, (a) where is the noise plot? The plot in figure (b) is skeptical, how can the posterior uncentainty be so low ($\sigma \approx 0.01$ K)? It is given as (see Rodgers 2000)

$$S_x = (S_a^{-1} + K^T S_\epsilon^{-1} K)^{-1}. \tag{1}$$

Is it verified? It can only happen in the case of almost perfect observations. It is difficult to say anything without having the knowledge about the observation and *a priori* errors used in the retrieval.

**Minor Comments:**

1) Page no. 5, line no. 138, What does "Kirchhoff-derived estimate" mean? Kirchoff's law is $r = 1 - \epsilon$. Both $r$ and $\epsilon$ are unknown. Is estimate of $r$ available from any other source?

2) Figure 13 (b), there are three plots in this figure, all shown in shades of gray. It is difficult to identify which is what. Authors should consider revising this figure for better presentation.

3) Figure 14 (d), I think line plots showing the mean along with error bars would be better. It requires binning in latitude and calculate the values (mean and standard deviation, or median and mean absolute deviation) in each bin.

4) I think it should be "Kelvin" not "kelvin", throughout the text.